# MULTI-RESOLUTION SKILLS FOR HRL AGENTS

## ABSTRACT

Hierarchical reinforcement learning depends on temporally abstract actions to solve long-horizon tasks. We propose Multi-Resolution Skills (MRS), a simple and scalable approach that constructs a discrete set of skill modules, each specialized to predict subgoals at a fixed temporal horizon (e.g., 8, 16, 32, 64 steps). Skill encoders share parameters, causing a minimal increase in model size while allowing each module to generate plans at a distinct temporal resolution. A learned meta-controller selects among these resolution-specific skills based on the task context; the meta-controller and skill policies are trained jointly with a single end-to-end objective in a single training phase. We evaluate MRS on DeepMind Control Suite, Gym-Robotics, and long-horizon AntMaze tasks. While maintaining computational efficiency, MRS consistently outperforms single-resolution baselines, yields meaningful gains over the HRL baselines in long-horizon navigation, and remains competitive with the non-hierarchical state-of-the-art (SOTA) on standard benchmarks. Ablations show that the multi-resolution design drives the improvement, suggesting temporal partitioning of skills is a useful inductive bias for HRL.

## 1 INTRODUCTION

Solving long-horizon control problems remains a central challenge in reinforcement learning: agents must plan across multiple time scales while retaining the ability to execute precise short-term maneuvers. Hierarchical reinforcement learning (HRL) addresses this by learning temporally abstract actions or skills that reduce planning complexity Ajay et al. (2021); Li et al. (2022); Sharma et al. (2020); Hafner et al. (2022). A common approach in prior work is to discover skills or subgoals by partitioning the state space, often via a learned latent distribution and unsupervised objectives that promote diversity or state coverage Eysenbach et al. (2019); Jiang et al. (2022); Sharma et al. (2020). Such methods implicitly mix temporal and geometric structure in the learned skill space, but do not explicitly provide options at different temporal horizons.

We illustrate the complementary behaviors produced by subgoals at different horizons using a simple 2D toy simulation (Fig. 1): nearby subgoals cause rapid corrective deviations, enabling precise steering, while longer-horizon subgoals produce smoother but less precise transitions. Motivated by this observation, we propose *Multi-Resolution Skills* (MRS), an HRL framework that explicitly partitions subgoals along temporal scales. MRS constructs a discrete set of skill modules, each specialized to produce subgoals at a fixed temporal distance (for example, 8, 16, 32, and 64 steps). To limit parameter growth, the skill encoders share a common backbone and differ only in their final layers; separate skill policies and a learned meta-controller select and execute the chosen subgoal. Crucially, the meta-controller and the per-resolution skill policies are trained jointly with a single end-to-end optimization objective, enabling the agent to learn which temporal resolutions are helpful for a given task and context. [Video: https://sites.google.com/view/multi-res-skills/home]

We implement MRS atop the Director Hafner et al. (2022), a SOTA HRL agent that provides a practical method for learning hierarchical behaviors directly from pixels, and evaluate on DeepMind Control Suite, Gym-Robotics, and long-horizon AntMaze navigation tasks. With minimal parameter overhead, MRS consistently outperforms single-resolution baselines, produces meaningful improvements over the Director baseline on long-horizon navigation tasks, and remains competitive with DreamerV3 Hafner et al. (2023) (non-HRL SOTA) on standard DeepMind Control Suite benchmarks.

Our contributions are as follows:

- We introduce Multi-Resolution Skills (MRS), a simple HRL design that explicitly partitions skills by fixed temporal horizons with minimal increase in model size (Sec. 3.2).

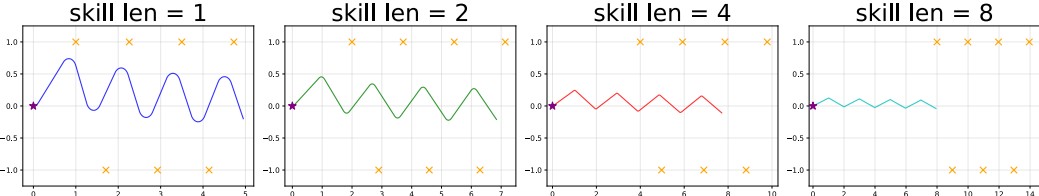

Figure 1: Simulation of a simple point agent (star) in a 2D grid that moves towards assigned goal positions (crosses). Goal updates every fixed number of steps $K$ and alternates between $(x + l_i, 1)$ and $(x + l_i, -1)$, where $x$ is the agent's current x-position and $l_i \in \{1, 2, 4, 8\}$ is the skill length. Goal positions impact agent behavior based on their distance from the agent state. Closer goals lead to more controlled and precise movements, but can be susceptible to incorrect goals. Meanwhile, far-away goals cause less deviation, leading to smooth but imprecise movements.

- We present a single-phase, end-to-end training procedure in which a learned meta-controller chooses among resolution-specific skill policies, enabling dynamic interleaving of fine- and coarse-grained actions (Sec. 3.3).
- We empirically evaluate MRS on continuous-control and long-horizon navigation benchmarks, and provide ablations that attribute gains to multi-resolution control (Secs. 5.1,5.2). MRS outperforms the Director at all tasks and matches DreamerV3 at most (which completely fails at some tasks).

## 2 BACKGROUND

### 2.1 DIRECTOR

The Director Hafner et al. (2022) is a recent SOTA model-based HRL agent composed of a world-model, worker, manager, and a Goal Variational AutoEncoder (VAE) Kingma et al. (2019). The world-model is implemented using the Recurrent State Space Module (RSSM) Hafner et al. (2019) that takes the environmental observations and constructs a state representation over time. The manager takes the state as input to yield a subgoal for the worker in the same state space (refreshed every $K$ steps). The worker takes the current state and the subgoal state to output an environmental action. The authors note that directly outputting subgoals for the worker in the state space by the manager results in a high-dimensional continuous control problem. Therefore, the Goal VAE learns a reduced categorical latent representation for the states, and the manager takes the current state as input to output a latent variable, which is expanded into a state using the Goal VAE decoder. The Goal VAE enables the manager to operate in the reduced latent space by facilitating the recall of states. We implement MRS using Director as the base architecture, modifying only the *manager policy* and the *Goal VAE*.

**Motivation**: It should be noted that the Goal VAE allows predicting states irrespective of the current state, which means the manager can select a goal $s_g$ unreachable by the worker. And by definition, the worker cannot reliably predict the right actions for unreachable goals. Therefore, given the current state, we propose constraining the search space to only nearby states, which can increase the search efficiency for appropriate goal states $s_g$. Furthermore, as mentioned in the Director paper and in our experiments with the Director, we observed that the worker rarely reaches the prescribed goal state $s_g$ within an episode. The manager only learns to select the goals $s_g$ so that they induce the proper actions from the worker that maximize the expected return, as evident in the manager's training objective. Rather than prescribing a goal state and waiting for the worker to reach it, we found that the manager assigns the worker goals as a moving target that the worker constantly chases. Thus, the goal states $s_g$ do not need to be strictly at the temporal length $K$ (the goal refresh rate). In fact, in our experiments with different temporal skill lengths $l$, we found that $l > K$ works significantly better than $l = K$ for our tasks (Sec. 5.2). We simulated a simple 2D point agent to follow goals prescribed at different distances, illustrating the behavioral differences (Fig. 1). Additionally, the appropriate skill length can be highly task-dependent (Fig. 7). Thus, we propose a Multi-Resolution Skills (MRS) mechanism that learns skills or abstract actions at multiple temporal resolutions and mixes them appropriately. Note that we use *temporal* to refer to the temporal distance of the assigned goal, not the duration for which it is executed.

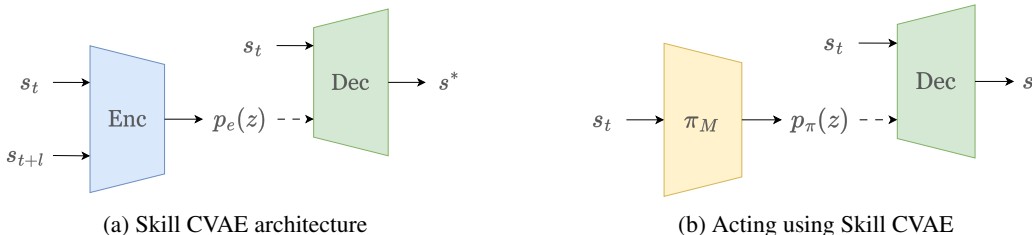

(a) Skill CVAE architecture            (b) Acting using Skill CVAE

Figure 2: Illustrations of the abstract state transition-based control for the manager. Dashed arrows indicate sample propagation from the predicted distribution. (a) Skill CVAE, where the Encoder encodes initial and final states $(s_t, s_{t+l})$ to a latent skill space and the Decoder reconstructs the final state using the initial state $s_t$ and a sampled skill variable. (b) The manager predicts the latent skills and then uses the Decoder to generate goals for the worker.

## 3 OUR METHOD

### 3.1 SKILLS AS ABSTRACT STATE TRANSITIONS

Given that the agent is at the state $s_t$, we want to constraint the goal predictions to states that can be achieved in $l$ steps. To do this, we replace the Goal VAE with a Conditional VAE (CVAE) that learns to predict possible future states $s_{t+l}$ conditioned on the current state $s_t$. The CVAE is learned online using the generated replay data. First, the replay trajectories are used to collect training examples as state pairs $(s_t, s_{t+l})$, where $s_{t+l}$ occurs $l$ steps after $s_t$. Then, the CVAE parameterized by weights $\phi$ is trained to optimize the ELBO objective (Eq. 1). It should be noted that it is the worker that predicts actions leading the agent to the goal state. CVAE is merely a skill recall mechanism that learns the abstract actions possible under the current worker policy and then allows the manager to modulate the worker's behavior predictably. Fig. 2 illustrates the skill-based architecture as a CVAE that learns skills online using the collected data (Fig. 2a). Fig. 2b shows how the manager can use the Skill CVAE during inference to generate sub-goals for the worker. Next, we present our method by scaling the concept of *skills* to multiple resolutions.

$$\mathcal{L}(\phi) = \|s_{t+l} - \text{Dec}_\phi(s_t, z)\|^2 + \beta\text{KL}[\text{Enc}_\phi(z|s_t, s_{t+l}) \| p(z)] \quad \text{where} \quad z \sim \text{Enc}_\phi(z|s_t, s_{t+l}) \tag{1}$$

### 3.2 MULTI-RESOLUTION SKILLS

Ideally, we want the manager to be able to predict any state that the worker can directly reach as a goal state. Instead of learning a single CVAE, we can learn multiple CVAEs, each specific to a temporal resolution. However, this can significantly increase the model size, thereby increasing the memory capacity and causing an unfair comparison. Thus, we keep all but the last layer of the encoder, and all but the first layer of the decoder, shared. The sharing causes a minimal increase in model size but increases the recall with the resolution-specific input and output layers. Fig. 3a illustrates the Multi-Resolution Skill CVAE architecture. For training, state-pairs $(s_t, s_{t+l_i})$ at $N$ different temporal resolutions $l_i \in \{l_0, l_1, ..., l_N\}$ are extracted from the replay data. Each training example is processed using the shared and the resolution-specific Encoder-Decoder layers. Then the total loss is calculated as the sum of the ELBO objectives of each CVAE and is optimized in a single step (Eq. 2) (the use of common layers is implied in the equations and is not mentioned to maintain simplicity). This results in the common layers being trained for all examples and the resolution-specific layers being trained only on the relevant examples. We use a mixture of 8, 8-dim categoricals ($8 \times 8$) as the prior distribution $p(z)$ for our CVAEs.

$$\mathcal{L}(\phi) = \sum_{i=0}^{N} \left\|s_{t+l_i} - \text{Dec}_\phi^i(s_t, z)\right\|^2 + \beta\text{KL}[\text{Enc}_\phi^i(z|s_t, s_{t+l_i}) \| p(z)] \quad \text{where} \quad z \sim \text{Enc}_\phi^i(z|s_t, s_{t+l_i}) \tag{2}$$

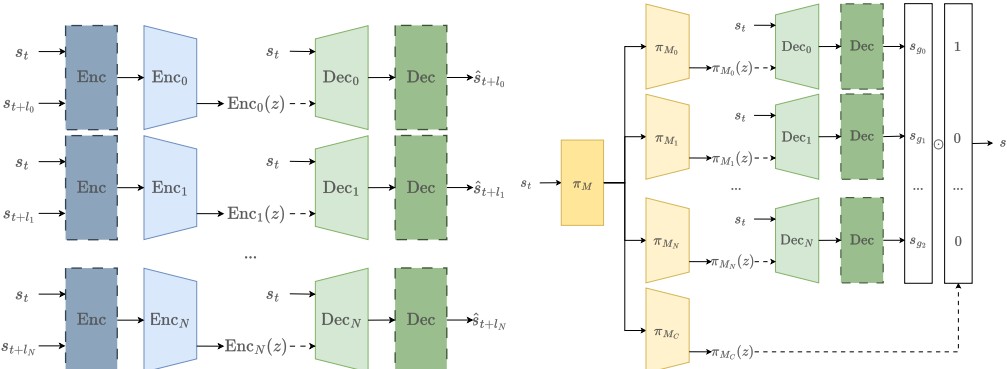

(a) Learning Multi-Resolution Skill CVAEs      (b) Acting using Multi-Resolution Skill CVAEs

Figure 3: Architectures for learning and acting using Multi-Resolution Skills ($l_i \in \{l_0, l_1, ..., l_N\}$). Dashed arrows indicate sample propagation from the predicted distribution. Dashed boundaries indicate shared layers. (a) Separate CVAEs are learnt for each temporal resolution $l_i$. The Enc and Dec modules represent the common layers of the Encoders and the Decoders, respectively. Each $\text{Enc}_i$ is the resolution-specific encoder output layer, and each $\text{Dec}_i$ is the resolution-specific decoder input layer. (b) The manager's policy has $N + 1$ output heads. $N$ skill heads $\pi_{M_i}$ that predict the resolution-specific skill latents and choice head $\pi_{M_C}$ that predicts an $N$-dimensional one-hot distribution. Samples from the skill latents are used to predict sug-goals using the respective Decoders, then the choice sample from $\pi_{M_C}$ selects one of the sub-goals as $s_g$ by gating.

## 3.3 MULTI-SKILL POLICY

The manager policy has $N + 1$ output heads, $N$ heads corresponding to each Skill CVAE that predicts latent distributions over skills $\pi_{M_i}(z|s_t)$, and an additional 'choice' head that predicts a one-hot $N$-dim distribution $\pi_{M_C}(c|s_t)$ (Fig. 3b). The latent skill samples are used to predict subgoals using the corresponding decoders (Eq. 3). And the one-hot choice sample selects from the subgoals by gating (Eq. 4). Fig. 3b illustrates the process of worker subgoal prediction using the Multi-Resolution Skill CVAEs. It should be noted that only the final layer of the policy is split into multiple heads, which does not increase the model capacity, but increases the recall capacity. The MRS policy is learned such that each skill head becomes an expert at using the corresponding resolution skills for all states $s_t \in \mathbf{S}$ independently. And the choice head simultaneously learns to pick the best skill head for all states $s_t \in \mathbf{S}$.

$$s_g^{i,t} = \text{Dec}_\phi^i(z_{t,i}, s_t) \quad \text{where} \quad z_{t,i} \sim \pi_{M_i}(z_{t,i}|s_t) \tag{3}$$

$$s_g^t = \sum_{i=0}^{N-1} c_{t,i}.s_g^{i,t} \quad \text{where} \quad c_t \sim \pi_{M_C}(c_t|s_t) \tag{4}$$

## 3.4 POLICY OPTIMIZATION

Like the Director Hafner et al. (2022), the MRS manager and the worker policies are implemented as Soft-Actor-Critics (SAC) and optimized using imagined trajectories. Imagination using the RSSM module helps cheaply generate on-policy data for training. The agent imagines a batch of $T$-step trajectories used to train both the manager and the worker. The returns are estimated using lambda returns, followed by policy update using policy gradients for the external and exploratory rewards. We briefly describe the common training steps below, followed by the exploratory objective (Sec. 3.4.1) and the policy gradients for our approach (Sec. 3.4.2). See Sec. B for full training and architecture details.

**Manager:** The manager is trained to maximize the external task and the exploratory rewards (Sec. 3.4.1). Since the manager works on a coarser temporal scale, an abstract trajectory of length $T/K$ is extracted, corresponding to every $K$-th step, and the rewards are summed within each non-overlapping subsequence of length $K$. Then, separate lambda returns are computed for each reward type, which

are learned using individual critics. The manager's policy is updated using policy gradients (3.4.2), using the weighted sum of advantages from both objectives.

**Worker:** The worker is trained to maximize the goal rewards, calculated as the cosine-max similarity between the agent state $s_t$ and the goal state $s_g$. The imagined trajectory is divided into $K$-step sub-trajectories, where the goal state $s_g$ remains constant. The rewards and lambda returns are computed for the sub-trajectories to update the critic, followed by policy update using the SAC objective.

### 3.4.1 Exploratory Loss

We aim to learn all possible abstract state transitions in the environment. Thus, we provide the manager with an additional exploratory reward that encourages it to discover novel state transitions. Since the Skills CVAE learns all possible abstract state transitions in the environment, we utilize the reconstruction error from the CVAE as a measure of novelty. This encourages the agent to repeat state transitions that are not yet well-learned by the CVAE. The exploratory reward $R_t^{\text{Expl}}(\tau)$ for the imagined trajectory $\tau$ of length $T$ is computed as the reconstruction error of the state $s_t$ conditioned on the starting state $s_0$ (Eq. 5). Since there are multiple CVAEs, we use the CVAE that best models the given state transition. Thus, the $\min$ of the reconstruction errors across all CVAEs is used as the reward.

$$R_t^{\text{Expl}} = \min_i \left\| s_t - \text{Dec}_\phi^i(s_0, z_{t,i}) \right\|^2 \quad \text{where} \quad z_{t,i} \sim \text{Enc}_\phi^i(z|s_0, s_t) \tag{5}$$

### 3.4.2 Policy Gradients For MRS

For clarity, we defer the full step-by-step derivation to Appendix A; here we summarize the main result and the training objectives used in practice. Under the MRS sampling protocol (skill latents $z_{k,i}$ from each skill head, a discrete choice $c_k$ selecting one head, and worker actions conditioned on the chosen subgoal), the trajectory log-probability decomposes into manager terms (choice and per-head latent policies) and worker terms. Applying the log-derivative trick and standard policy-gradient manipulations (with temporal abstraction indexed by abstract step $k$ with refresh interval $K$) yields the manager policy gradient of the form:

$$\nabla_M J = \mathbb{E}_\tau \left[ \sum_{k=0}^{\lfloor T/K \rfloor - 1} \left( \underbrace{\nabla_M \log \pi_{M_C}(c_k \mid s_{kK})}_{\text{Choice head}} + \sum_{i=0}^{N-1} \underbrace{c_{k,i} \nabla_M \log \pi_{M_i}(z_{k,i} \mid s_{kK})}_{\text{Skill head } i} \right) \left( G_k^\lambda - v_M(s_{kK}) \right) \right]$$

Where $G_k^\lambda$ are lambda-returns computed over abstract steps and $v_M$ is the manager critic. In practice, each manager head (choice head and per-resolution skill heads) is optimized with an actor loss using the above advantage estimator plus an entropy regularizer, and the critic is trained with a squared-error target on $G_k^\lambda$ (see Appendix B for exact loss definitions, variance-reduction details, and implementation notes). This compact formulation makes it clear that the manager gradient separates into a choice term and per-head terms (the $c_{k,i}$ factor for the skill head gradients effectively selects the active head), enabling joint, single-phase end-to-end optimization of the meta-controller and resolution-specific skill policies.

## 4 Addressing A Critical Failure Mode

Previous skill discovery methods have mentioned difficulties learning skill primitives while acting using the same skills Hafner et al. (2022); Eysenbach et al. (2019). This is because, after the model learns a few reliable skills, it tends to repeat them, thereby getting stuck with suboptimal skills. This happens if the CVAEs prematurely converge before the policy; one way to force this is to increase the training data for the CVAE disproportionately. This causes the policy to collapse into a degenerate solution, as the CAVE predicts only the initially learned subgoals. For example, the quadruped embodiment can learn to stand, but freezes thereafter. For completeness of the solution, we make an additional modification that prevents this problem. In addition to the Skill CVAEs, we introduce another VAE that learns states unconditionally, similar to the Director. The unconditional VAE $(\text{Enc}_\psi^\infty, \text{Dec}_\psi^\infty)$ predicts subgoal states $s_g$ completely independent of any previous state $s_t$, imitating

learning $\infty$-length skills. This helps the agent escape the collapse by allowing the manager to select temporally unconstrained goal states. Thus, if the skills CVAEs have collapsed, the manager can use this VAE to generate subgoals that the skill CVAE cannot yet, removing any need to balance policy and CVAE learning. Our results show that the agent initially uses the unconditional VAE but soon switches to Skill CVAEs (Fig. 5).

## 5 RESULTS

For generality, we use skill lengths $L = [64, 32, 16, 8, \infty]$ for all our experiments and keep the rest of the hyperparameters the same as the base architecture. Since we use multiple policy heads, the policy learning signal is split between the $N$ skill heads and diluted by a factor of $N (= 5)$; thus, we increase training to every 8-th step rather than 16 (for MRS and the Director). The above configuration ensures that all performance changes are strictly due to the proposed architectural changes only (which allow recall of subgoal states at multiple temporal resolutions). We use a similar-sized DreamerV3 that trains every 2-nd step, making it 4 times more expensive.

We first evaluate the agent in a variety of standard benchmarks, including the DeepMind Control Suite and Gym-robotics tasks, to test its performance against the Director Hafner et al. (2022) (SOTA HRL) and DreamerV3 Hafner et al. (2023) (SOTA non-HRL) agents. Then, the agent is evaluated in long-horizon AntMaze tasks with sparse rewards to test whether the agent can learn solely using the exploratory objective. Finally, we conduct ablation studies to test the impact of dynamic interleaving of skills and also compare against skill discovery methods that also learn abstract actions without external rewards.

### 5.1 STANDARD BENCHMARKS

We compare our method with SOTA methods (Director Hafner et al. (2022) and DreamerV3 Hafner et al. (2023)) on several tasks in the DeepMind Control Suite (DMC) Tassa et al. (2018) and Gymnasium-Robotics de Lazcano et al. (2024).

**DeepMind Control Suite**: For DMC, each episode lasts for 1000 steps before terminating and provides positive dense rewards at each step. Fig. 4 shows the performance of our method compared to the baselines. The results show that our method outperforms Director at all tasks and matches DreamerV3's performance in most cases. Thus, MRS closes the performance gap between Director and DreamerV3 while retaining the compute efficiency of Director and a similar model size. We also plot the evolution of the choice distribution during training (Fig. 5). A common trend in some tasks was that the manager initially preferred unconditional VAE, but later switched to skill CVAEs (Fig. 5). This trend is similar to human behavior when learning new skills, e.g., body movements for a new sport. Initially, one might make crooked motions through a few identified advantageous body configurations, but repetitions develop skills and reduce future conscious effort Sanes (2003).

**Gym Robotics**: For Robotic tasks (*Push* and *Pick n Place*), each episode lasts for 100 steps and incurs an existence penalty of $-1$ each step. Thus, the rewards are sparse, but the task is of a much shorter horizon than the AntMaze. It can be seen that DreamerV3 completely fails at the task while Director and MRS perform well (Fig. 4). MRS edges out the Director slightly in terms of performance and convergence speed.

**Egocentric Ant**: We also tested our method on the Egocentric ant maze task, where the agent receives sparse rewards for reaching a goal location (1 on success and 0 otherwise). Each episode lasts 3000 steps before terminating. Therefore, training is mainly done using exploratory rewards. The agent takes the proprioceptive observations and an egocentric camera image as inputs. While DreamerV2 fails at the task, the Director and MRS solve it, with MRS receiving higher scores. This task takes extremely long to complete, so we take results from Hafner et al. (2022) for comparison (Fig. 6).

### 5.2 ABLATIONS

*How well do the individual skills perform, and is the dynamic skill interleaving useful?*

Our method trains individual expert policies for each skill CVAE, for all states $s_t \in \mathbf{S}$, and the choice head for selecting the best skill for all states $s_t \in \mathbf{S}$. In this context, we compare the following settings: the default choice mechanism, random choice, and using each skill module separately. The skill selection mechanism is modified in an already trained MRS agent to enforce the above settings.

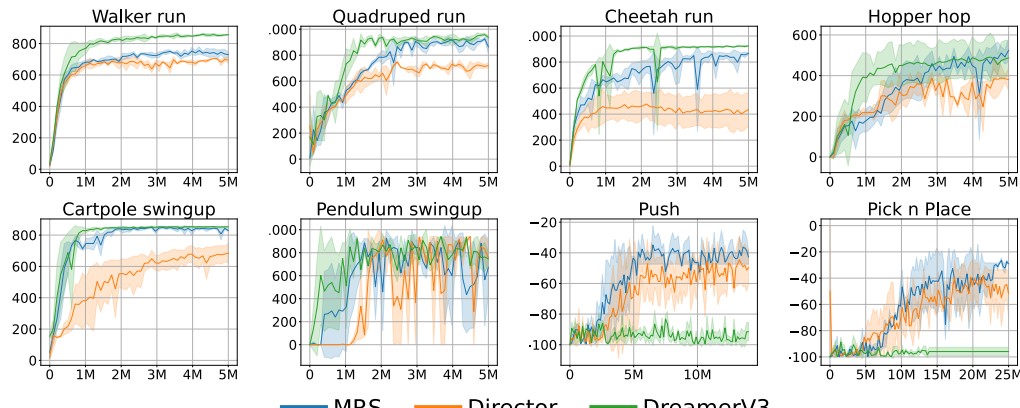

Figure 4: Episode scores from MRS (ours), Director, and DreamerV3 (3 seeds per experiment). The plot shows the total rewards (mean and standard deviation) received in an episode against the environmental step. Both methods use the same common hyperparameters. The first *six* tasks are from the DMC suite and the last *two* (Push and Pick n Place) are from the Gymnasium-Robotics suite. It can be seen that MRS boosts performance of the base model noticiably in all cases while maintaining the compute efficiency.

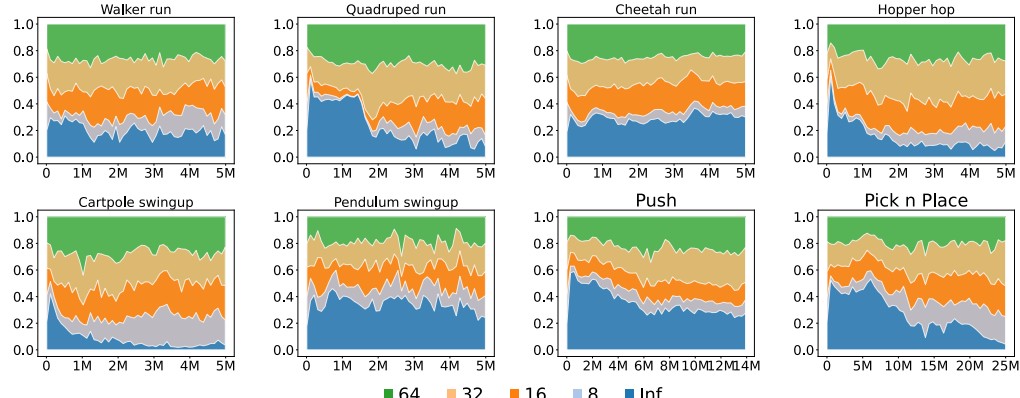

Figure 5: Stream graphs showing the evolution of the choice distribution during training averaged across 3 seeds. A trend can be observed in some tasks where the manager initially focuses on $\infty$-length skills but gradually shifts to temporally constrained skills.

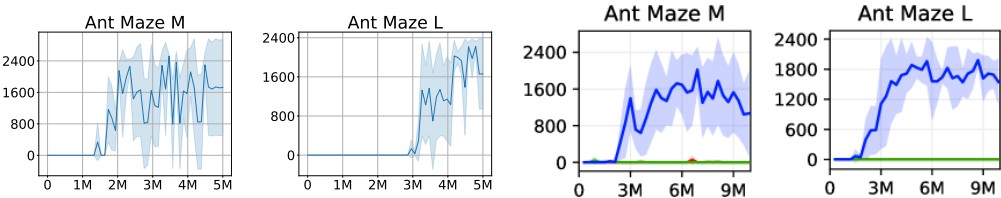

Figure 6: Episode rewards from the Egocentric Ant Maze task against the environmental step during training (3 seeds). (Left) — MRS (*Ours*), (Right) Results taken from Hafner et al. (2022) that compares: — Director, — Director with worker receiving external task reward, — DreamerV2. It can be noticed that MRS reaches the peak performance ($\sim 2400$) at around 3.5M steps (medium) and 5M steps (large), while the Director reaches peack performance of $\sim 1800$ at 7M and 6M steps respectively.

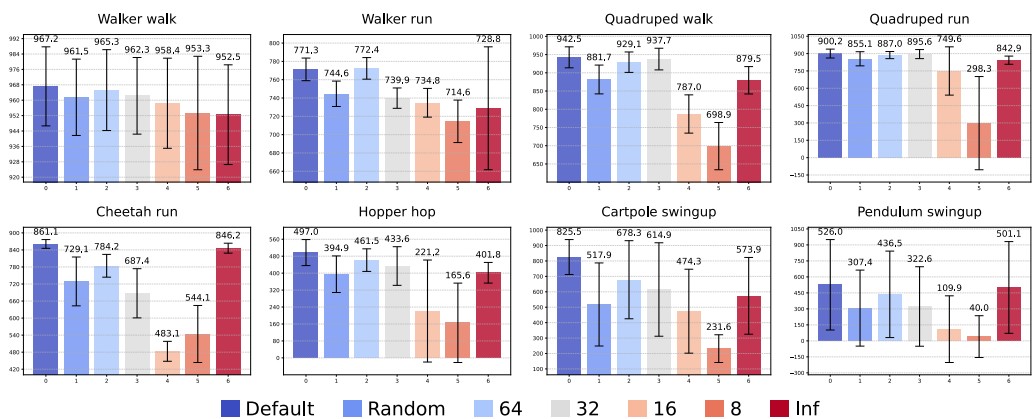

Figure 7: Final performance comparison between the settings: default choice mechanism, random selection, and the skills individually $[64, 32, 16, 8, \infty]$. The results are the mean and standard deviations of the episodic rewards across 100 evaluation runs.

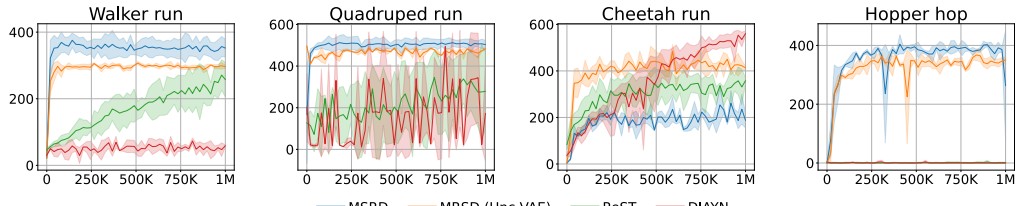

Figure 8: Performance comparison of agents fine-tuned for tasks after an exploration phase (3 seeds per experiment). The graphs show total episodic rewards (mean and standard deviation) against the global steps. The plots compare: MRS, MRS using exploratory rewards from the unconditional VAE, ReST, and DIAYN. Our agent is trained every 8-th step using image inputs, while DIAYN/ReST trains every step using the internal environmental proprioceptive state.

Fig. 7 shows the results for some DMC tasks where each skill score is averaged across 100 episodes. It can be seen that interleaving the skills using the proposed choice mechanism consistently yields the best results. It should also be seen that no individual skill performs well for all tasks; thus, using the choice policy $\pi_{M_C}$ can help automate skill selection. Another notable fact was that while the agent prefers the $\infty$-length skills for the `cheetah_run` task (Fig. 5), the Director agent that only uses $\infty$-length skills fails to perform (Fig. 4). Indicating necessity of multi-resolution skill interleaving.

*Can the agent learn usable skills only using the exploratory objective?*

We test whether MRS can learn skills independently of external rewards and then learn a policy to utilize these skills to perform a task. For this, we first train an agent for 3M environmental steps using only the exploratory objective. The agent learns interesting behaviors, such as backflips, headstands, somersaults (both forward and backward), etc. (Appendix C). Next, keeping all modules static, we fine-tune the manager policy and a fresh critic for the environmental task rewards for 1M environmental steps. We compare our method to two previous skills discovery methods: ReST Jiang et al. (2022) and DIAYN Eysenbach et al. (2019). Both methods maximize an information-theoretic objective to learn a set of distinct skills. Then, the skill that yields the maximum rewards for the external task is further fine-tuned. The original results on the methods are at the `Gym` embodiments of the same agents, so we use their respective parameters, including reward scaling. We also compare our exploratory objective against the Director's, which is computed as the reconstruction error using the unconditional VAE (Sec. 4). Fig. 8 shows the comparisons. It can be seen that our method performs well for all tasks except the `cheetah_run`, while other methods struggle to do so.

*Are there any qualitative differences between states where certain skills are preferred over others?*

We segregate states in a trajectory by the choice variable to manually verify if certain skill-lengths are consistently preferred over others in various situations (Sec. G). For example, the walker agent prefers 8-step skills either when the agent has both feet on the ground or in a fully extended stance,

and prefers 64-length skills in a mid-lunge stance (Fig. 18). We also observe that the agents prefer $\infty$-length skills for less frequently visited states, such as being dropped at the episode start or when it mistakenly topples mid-episode. While we do not draw any parallels with the human running gait, it is clear that certain skills are preferred in certain situations.

## 6 RELATED WORK

**Hierarchical RL and options.** Hierarchical reinforcement learning (HRL) formalizes temporal abstraction by allowing agents to select temporally-extended actions or *options* instead of only primitive actions Sutton et al. (1999); Barto and Mahadevan (2003); Botvinick et al. (2009); Wiering and Van Otterlo (2012); Pateria et al. (2021) Early ideas of feudal/manager–worker decomposition date back to Dayan and Hinton (1992); more recent neural instantiations include FeUdal Networks Vezhnevets et al. (2017) and Options-Critic style approaches that learn option-policies and termination conditions end-to-end Bacon et al. (2017). Manager–worker schemes have also been applied with goal-conditioned workers (e.g., HIRO) where a higher-level policy proposes subgoals and a lower-level controller is trained to reach them Nachum et al. (2018).

**Unsupervised skill discovery via mutual information.** A large body of work focuses on unsupervised discovery of diverse skills by maximizing information between a latent skill variable and state trajectories or state pairs. Representative methods include DIAYN Eysenbach et al. (2019), DADS Sharma et al. (2020), InfoGAN-based approaches Kurutach et al. (2018), and OPAL Ajay et al. (2021); these methods differ in whether they maximize MI with single states, state pairs, or whole trajectories, and whether the learned skills are later used for planning or exploration. Variants such as ReST train skills sequentially to increase coverage Jiang et al. (2022). The principal advantage of these methods is their broad behavioral diversity without the need for external rewards; however, they do not provide an explicit mechanism to select temporal resolutions for subgoals on their own.

**Goal-conditioned, model-based and hybrid HRL.** More recent research combines learned skills or goal representations with model-based planning or learned world models to improve long-horizon performance Hafner et al. (2022); Li et al. (2022). These works demonstrate that compact goal representations and explicit subgoal prediction can facilitate planning; however, they typically do not explicitly partition skills into fixed temporal horizons. Our Multi-Resolution Skills (MRS) approach complements these lines by explicitly training resolution-specific skill heads (fixed temporal distances) and a learned meta-controller that selects among them in a single end-to-end phase; in this sense, MRS sits between unsupervised options discovery and goal-conditioned manager–worker HRL.

## 7 DISCUSSION & FUTURE WORK

We propose a novel skill discovery framework that explicitly partitions the state space by temporal resolution, enabling hierarchical control through multi-resolution skill modules. Our agent outperforms Director with minimal architectural changes and achieves performance parity with DreamerV3 on standard benchmarks with better training efficiency.

The key findings that emerge from our analysis are:

- **HRL Bottleneck**: Limited recall capacity for options can be a bottleneck for HRL agents. Simply increasing the number of distinct executable options available can increase performance 4.
- **Skill Interleaving Matters**: Ablation studies show that the skill-interleaving agent performs the best, and no single skill works best across all tasks (Fig. 7).
- **Reward-Agnostic Learning**: The agent successfully discovers usable skills without external rewards through latent space exploration (Figs. 6,8). We see a small limitation when our exploration rewards do not perform as well as the others at the `cheetah_run` task (Fig. 4), indicating that no single reward scheme is sufficient for all tasks.

The architecture is highly flexible, allowing for the mixing of learned and deterministic skills, resulting in hybrid structures. The skills can also be used as abstract actions for goal-directed motion planning. The multi-head policy gradient formulation can also be easily extended to other RL algorithms.

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

# A    DERIVATION OF POLICY GRADIENTS FOR MRS

We first decompose the action prediction process to derive the policy gradient to train the manager and worker policies. Let an MRS agent be in state $s_t$ at step $t$. Every $K$-th step, the manager refreshes the worker's goal. For clarity, let the abstract step be indexed by $k$, then at each abstract step ($t = kK$):

1. Sample skill latents from the skill heads: $z_{k,0}, z_{k,1}, ..., z_{k,N-1} \sim \Pi_{i=0}^{N-1} \pi_{M_i}(z_{k,i}|s_{kK})$.

2. Sample a choice variable: $c_k \sim \pi_{M_C}(c_k|s_{kK})$.

3. Compute the selected subgoal: $s_g^k = \sum_{i=0}^{N-1} c_{k,i} \cdot \text{Dec}_\phi^i(s_{kK}, z_{k,i})$.

4. Predict the environmental actions using worker: $\pi_W(a_t|s_t, s_g^k)$

Thus, the trajectory probability that starts at $s_0$ can be written as:

$$p(\tau) = p(s_0) \prod_{k=0}^{\lfloor T/K \rfloor - 1} \pi_{M_C}(c_k|s_{kK}) \underbrace{\prod_{i=0}^{N-1} \pi_{M_i}(z_{k,i}|s_{kK})^{c_{k,i}}}_{\text{Manager}} \prod_{t=0}^{T-1} \underbrace{\pi_W(a_t|s_t, s_g^{\lfloor t/K \rfloor})}_{\text{Worker}} \cdot \underbrace{p_T(s_{t+1}|a_t, s_t)}_{\text{State transition}}$$

$$(6)$$

The components of the equation can be read as: the manager predicts the skills $(z_{k,0}, z_{k,1}, ..., z_{k,N})$ and choice $c_k$ for every abstract step $k$, the worker predicts the action $a_t$ at each step $t$ using the subgoal $s_g^{\lfloor t/K \rfloor}$ for the duration, and the environmental state transition $p_T$. Here, the exponent $c_{k,i}$ collapses the skill probabilities $\pi_{M_i}(z_{k,i}|s_{kK})$ of the unselected skill head to 1 as they do not affect the trajectory.

We follow the policy gradient derivation from Sutton and Barto (2018). The aim is to compute $\nabla_\theta J$, where $J = \mathbb{E}_\tau[R(\tau)]$ is the expected reward and $\theta$ are the policy parameters. Using the standard log-derivative trick (Sutton and Barto (2018)), the objective can be written as maximizing the trajectory log-probability weighted by the expected reward:

$$\nabla_\theta J = \mathbb{E}_\tau[R(\tau) \cdot \nabla_\theta \log p(\tau)]$$

The gradient of the trajectory log-probability w.r.t. the manager parameters $M$ is:

$$\nabla_M \log p(\tau) = \sum_{k=0}^{\lfloor T/K \rfloor - 1} [\nabla_M \log \pi_{M_C}(c_k|s_{kK}) + \sum_{i=0}^{N-1} c_{k,i} \nabla_M \log \pi_{M_i}(z_{k,i}|s_{kK})]$$

Therefore, the policy-gradient objective can be written as:

$$\nabla_M J = \mathbf{E}_\tau[R(\tau) \cdot \sum_{k=0}^{\lfloor T/K \rfloor - 1} [\nabla_M \log \pi_{M_C}(c_k|s_{kK}) + \sum_{i=0}^{N-1} c_{k,i} \nabla_M \log \pi_{M_i}(z_{k,i}|s_{kK})]]$$

Given these policy gradients, we construct the losses for each head as the sum of the policy gradient objective and an entropy maximization objective (Eq. 9,8), and sum them for the total loss (Eq. 10).

$$G_k^\lambda = R_k + \gamma((1-\lambda)v_M(s_{kK}) + \lambda G_{k+1}^\lambda) \tag{7}$$

$$\mathcal{L}(\pi_{M_c}) = -\mathbb{E}_\tau \sum_{k=0}^{\lfloor T/K \rfloor - 1} \log \pi_{M_c}(c_k|s_{kK})(G_k^\lambda - v_M(s_{kK})) + \eta \mathbb{H}[\pi_{M_C}(c_k|s_{kK})] \tag{8}$$

$$\mathcal{L}(\pi_{M_i}) = -\mathbb{E}_\tau \sum_{k=0}^{\lfloor T/K \rfloor - 1} c_{k,i} \cdot \log \pi_{M_i}(z_{k,i}|s_{kK})(G_k^\lambda - v_M(s_{kK})) + \eta \mathbb{H}[\pi_{M_i}(z_{k,i}|s_{kK})] \tag{9}$$

$$\mathcal{L}(\pi_M) = \mathcal{L}(\pi_{M_c}) + \sum_{i=0}^{N-1} \mathcal{L}(\pi_{M_i}) \tag{10}$$

$$\mathcal{L}(v_M) = \mathbb{E}_\tau \sum_{k=0}^{\lfloor T/K \rfloor - 1} (v_M(s_{kK}) - G_k^\lambda)^2 \tag{11}$$

Where $G_k^\lambda$ is the lambda returns estimated using abstract trajectories (Eq. 7), $v_M$ is the critic (Eq. 11). The policy maximizes the advantage $G_k^\lambda - v_M(s_{kK})$ instead of directly maximizing estimated rewards. Weighted entropic losses $\mathbb{H}[\cdot]$ encourage adequate exploration prior to convergence. The manager learns separate critics, and estimates separate returns and advantages for the external and exploratory rewards. And the total advantage is the weighted sum of exploratory and external advantages ($([1.0, 0.1])$ for our case).

# B  ARCHITECTURE & TRAINING DETAILS

## B.1  WORKER

The worker is trained using $K$-step imagined rollouts ($\kappa \sim \pi_W$). Given the imagined trajectory $\kappa$, the rewards for the worker $R_t^W$ are computed as the `cosine_max` similarity measure between the trajectory states $s_t$ and the prescribed worker goal $s_{wg}$. First, discounted returns $G_t^\lambda$ are computed as $n$-step lambda returns (Eq. 12). Then the Actor policy is trained using the SAC objective (Eq. 13) and the Critic is trained to predict the discounted returns (Eq. 14). The entropy for the worker and the manager is weighted to maintain a target entropy.

$$G_t^\lambda = R_{t+1}^W + \gamma_L((1-\lambda)v(s_{t+1}) + \lambda G_{t+1}^\lambda) \tag{12}$$

$$\mathcal{L}(\pi_W) = -\mathbb{E}_{\kappa \sim \pi_W} \sum_{t=0}^{H-1} \left[ (G_t^\lambda - v_W(s_t)) \ln \pi_W(z|s_t) + \eta \mathrm{H}[\pi_W(z|s_t)] \right] \tag{13}$$

$$\mathcal{L}(v_W) = \mathbb{E}_{\kappa \sim \pi_W} \left[ \sum_{t=0}^{H-1} (v_W(s_t) - G_t^\lambda)^2 \right] \tag{14}$$

## B.2  IMPLEMENTATION DETAILS

We implement two functions: `policy` (Alg. 2) and `train` 1, using the hyperparameters shown in Table 1. The functions are implemented in Python/Tensorflow using XLA JIT compilation. The experiments on average take 2 days to run 5M steps on an NVIDIA RTX 5000.

| Name | Symbol | Value |
|---|---|---|
| Train batch size | $B$ | 16 |
| Replay data length | - | 64 |
| Worker abstraction length | $K$ | 8 |
| Explorer Imagination Horizon | $T$ | 16 |
| Return Lambda | $\lambda$ | 0.95 |
| Return Discount | $\gamma$ | 0.99 |
| Skill resolutions | $L$ | $\{64, 32, 16, 8, \infty\}$ |
| Target entropy | $\eta$ | 0.5 |
| KL loss weight | $\beta$ | 1.0 |
| RSSM deter size | - | 1024 |
| RSSM stoch size | - | $32 \times 32$ |
| Optimizer | - | Adam |
| Learning rate (all) | - | $10^{-4}$ |
| Adam Epsilon | - | $10^{-6}$ |
| Weight decay (all) | - | $10^{-2}$ |
| Activations | - | LayerNorm + ELU |
| MLP sizes | - | $4 \times 512$ |
| Train every | - | 8 |
| Prallel Envs | - | 4 |

Table 1: Agent Hyperparameters

---

**Algorithm 1:** Multi-Resolution Skill Training

---

**Input:** Collected trajectories $\mathcal{D} = \{\tau_1, ..., \tau_B\}$
**Output:** Updated world model wm, skill modules $(\text{Enc}_\phi, \text{Dec}_\phi)$, manager $\pi_M$, worker $\pi_W$

```
// World Model Training
wm.train(D)                                    // See Hafner et al. (2019)
// Multi-Resolution Skill Learning
L_skills ← [ ]
for l_i ∈ L do
    {(s_t, s_{t+l_i})} ← ExtractStatePairs(D, l_i)
    L_i ← skill_loss(s_t, s_{t+l_i})           // CVAE loss (Eq. 1)
    L_skills.append(L_i)
update_skills(sum(L_skills))

// Policy Optimization via Imagination
S_init ← {s_0 | s_0 ∈ τ, τ ∈ D}               // Initial states
τ̂ ← wm.imagine(π_MRS, S_init, T)   // Rollout imagined trajectories (Alg.
 2)

// Reward Computation
τ̂.r^extr ← r_env(τ̂)                           // Environment reward
τ̂.r^expl ← expl_rew(τ̂)                        // Exploration reward (Eq. 5)
τ̂.r^goal ← cosine_max(τ̂.s_t, τ̂.s_g^⌊t/K⌋)     // Goal achievement reward

// Hierarchical Policy Update
T_W ← split(τ̂)                                // Worker-level transitions
T_M ← abstract(τ̂)                             // Manager-level abstractions
L(π_M), L(v_M) = manager_loss(T_M)                      // Eqs. 10,11
update_manager(L(π_M), L(v_M))
L(π_W), L(v_W) = worker_loss(T_W)                       // Eqs. 13,14
update_worker(L(π_W), L(v_W))
```

---

---

**Algorithm 2:** Multi-Resolution Skill Policy ($\pi_{\text{MRS}}$)

---

**Input:** Observation $o_t$, Agent state $(t, s_{t-1}, a_{t-1}, s_g)$
**Output:** Action $a_t$, New agent state $(t+1, s_t, a_t, s_g)$

---

$s_t \leftarrow \text{wm}(o_t, s_{t-1}, a_{t-1})$          // World model state update

**if** $t \bmod K = 0$ **then**
   |   // Manager updates goal every $K$ steps
   |   $(z_0, z_1, ..., z_{N-1}, c) \sim \pi_M(s_t)$     // Sample skill latent $z$ and choice $c$
   |   $\{s_g^i\}_{i=0}^{N-1} \leftarrow \{\text{Dec}_\phi^i(s_t, z_i)\}_{i=0}^{N-1}$       // Generate candidate goals
   |   $s_g \leftarrow \sum_{i=0}^{N-1} c_i \cdot s_g^i$        // Select goal using choice vector $c$
**else**
   |   $s_g \leftarrow s_g$                    // Persist previous goal

// Worker policy execution
$a_t \leftarrow \text{Worker}_\pi(s_t, s_g)$       // Generate action for current goal

Return $a_t, (t+1, s_t, a_t, s_g)$

---

## C   BEHAVIORS LEARNED VIA EXPLORATION

We observed some interesting behaviors that the MRS agent regularly exhibited, such as front flips, back flips, and jumps, while training solely with the exploratory loss. The intrinsic exploratory loss encourages the agent to perform novel state transitions (Sec. 3.4.1). Fig. 9 shows some of the learned movements.

## D   BROADER IMPACTS

### D.1   POSITIVE IMPACTS

Our method's sample efficiency (training every 8 steps) could reduce computational costs for real-world robot training, thereby lowering environmental footprints. The imagination-based policy optimization mitigates hazards that can occur during learning. The skill-interleaving mechanism enables transparent agents with interpretable subgoals. The learned skills can be interleaved with rigorously tested safe skills, and the selection can be appropriately constrained to mitigate failures.

### D.2   NEGATIVE IMPACTS AND MITIGATIONS

- **Inaccurate Training**: Imagination can cause incorrect learning. Mitigation: Rigorous testing using manual verification of world-model reconstructions against ground truths.

- **Malicious Use**: Hierarchical control could enable more autonomous adversarial agents. Mitigation: Advocate for gated release of policy checkpoints.

### D.3   LIMITATIONS OF SCOPE

Our experiments focus on simulated tasks that do not involve human interaction. Real-world impacts require further study of reward alignment and failure modes.

## E   LLM USAGE

We used LLMs to refine the abstract, introduction, and background sections of our paper, primarily to polish the language.

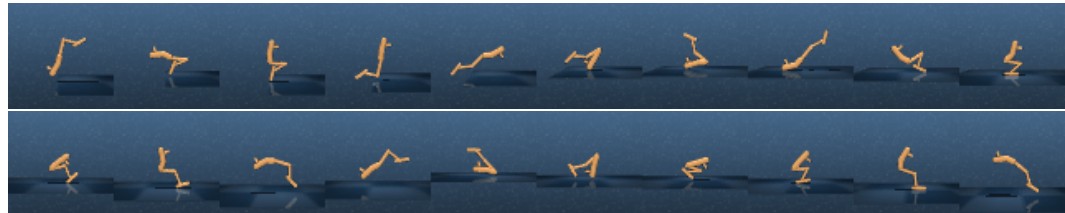

(a) Hopper learns to use a front flip to stand, and back flips.

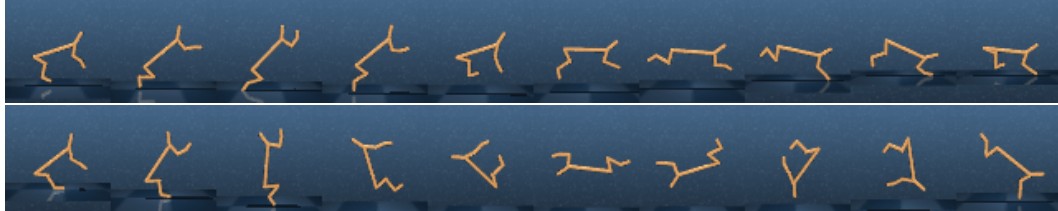

(b) Cheetah learns to leap forward and perform perfect back flips.

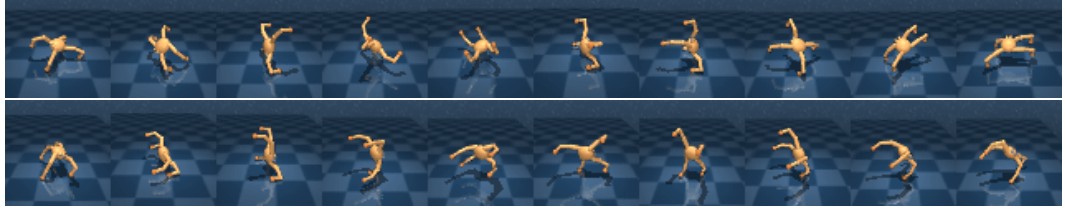

(c) Quadruped learning side rolls and walking on two legs.

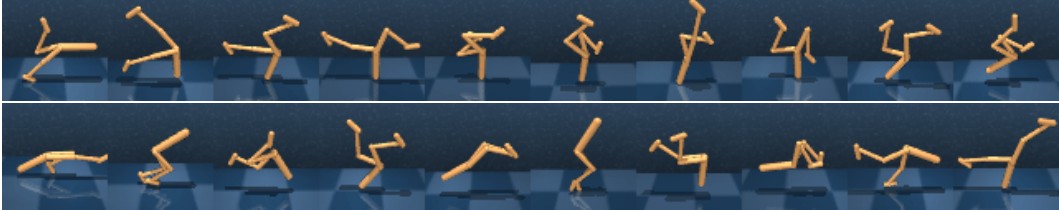

(d) Walker trying to headstand repeatedly and fast-forward tumbling using head and legs.

Figure 9: Samples of some movements learned and regularly performed by the agent optimized only for the exploratory loss.

# F  SAMPLE GOALS USING SKILL CVAES

We generate some sample goals using each of the Skill CVAEs individually. The goals are generated using a uniform prior $p(z)$ for the skills, and initial states $s_t$ sampled from the replay database. We use skills with temporal resolutions $[64, 32, 16, 8, \infty]$, but we omit the $\infty$ length skills, as they correspond to simply learning all states independently and are not our contribution.

## F.1  DEFAULT OBJECTIVE

The agent is trained using the default objective (weighted external and exploratory advantages). Since we use a strong bias towards external reward ($[1.0, 0.1]$), the skills learned are biased towards the goal states more appropriate for the objective. We sample the goals for tasks: `walker_run` (Fig. 10), `quadruped_run` (Fig. 11), `cheetah_run` (Fig. 12), and `hopper_hop` (Fig. 13).

## F.2  EXPLORATION ONLY

The agent is optimized only for the exploration objective, which aims to maximize coverage of the state transition space. We sample the goals per Skill CVAE for embodiments: `walker` (Fig. 14), `quadruped` (Fig. 15), `cheetah` (Fig. 16), and `hopper` (Fig. 17) in the DMC suite Tassa et al. (2018).

# G  VISUALIZING CHOICE PREFERENCES FOR STATES

To check if the agent exhibits any choice preferences for states, we run the agent for 5 episodes. Then the states $s_t$ visited by the agent are segregated by the choice ($c_t$) made by the agent and shown below. The visualizations can help verify if there are any correlations between the agent state $s_t$ and the choice variable $s_t$. We sample the states for the tasks: `walker_run` (Fig. 18), `quadruped_run` (Fig. 19), `cheetah_run` (Fig. 20), and `hopper_hop` (Fig. 21).

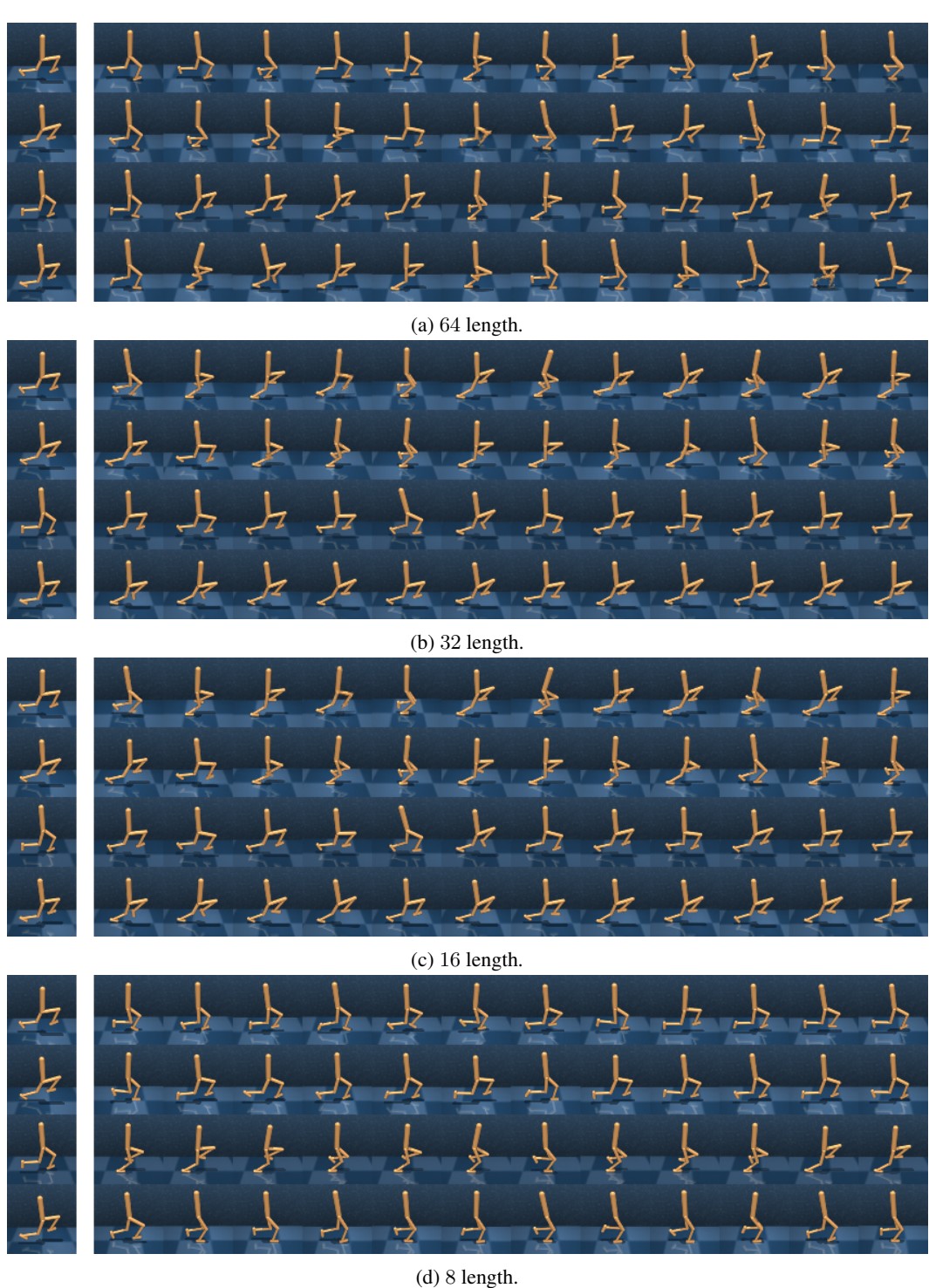

(a) 64 length.

(b) 32 length.

(c) 16 length.

(d) 8 length.

Figure 10: Sample goals from the `walker_run` task learned using the external and the exploratory rewards. The images on the left show the current state $s_t$, and the remaining images show the goal options generated by different skill CVAEs.

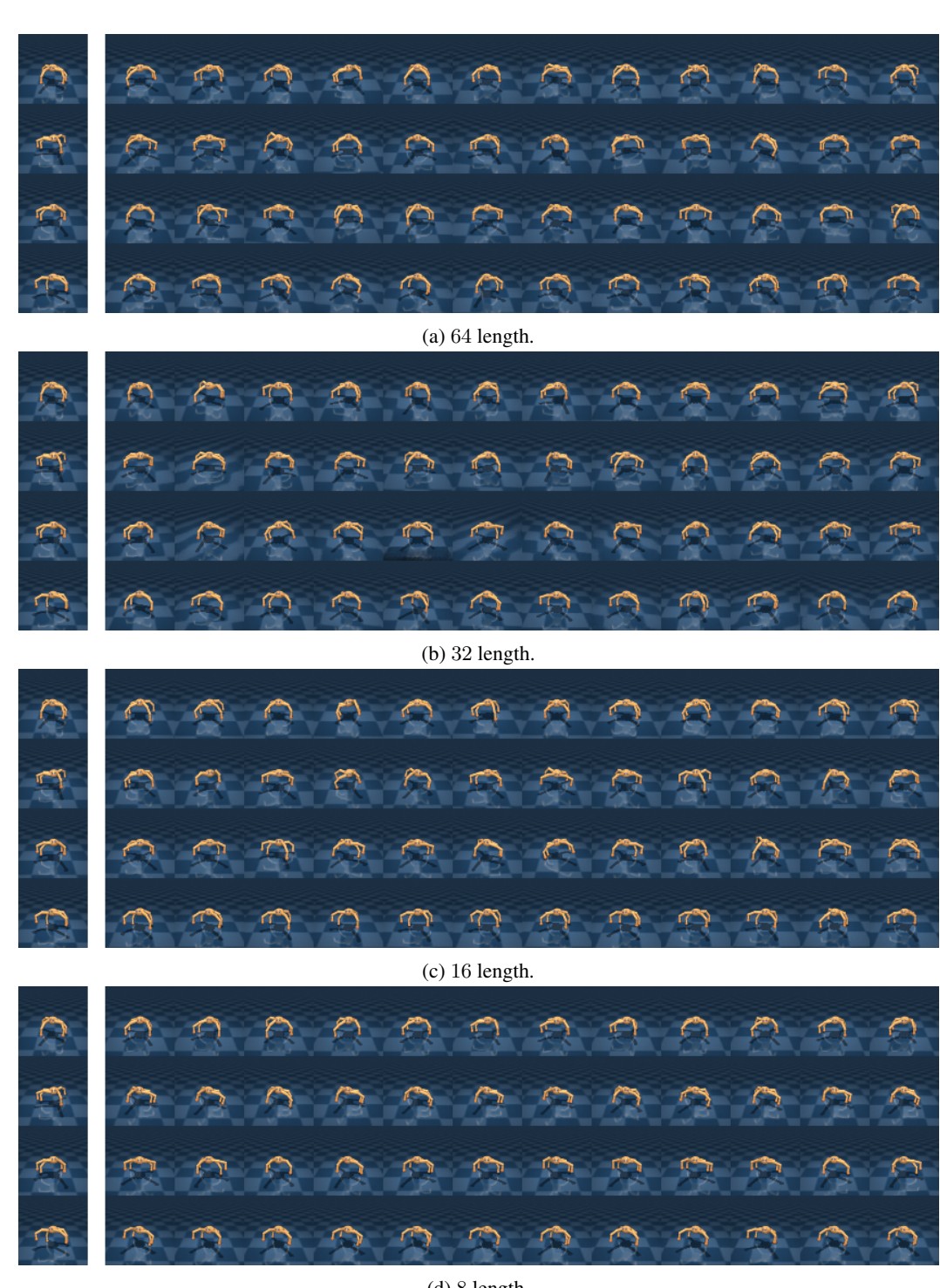

(a) 64 length.

(b) 32 length.

(c) 16 length.

(d) 8 length.

Figure 11: Sample goals from the `quadruped_run` task.

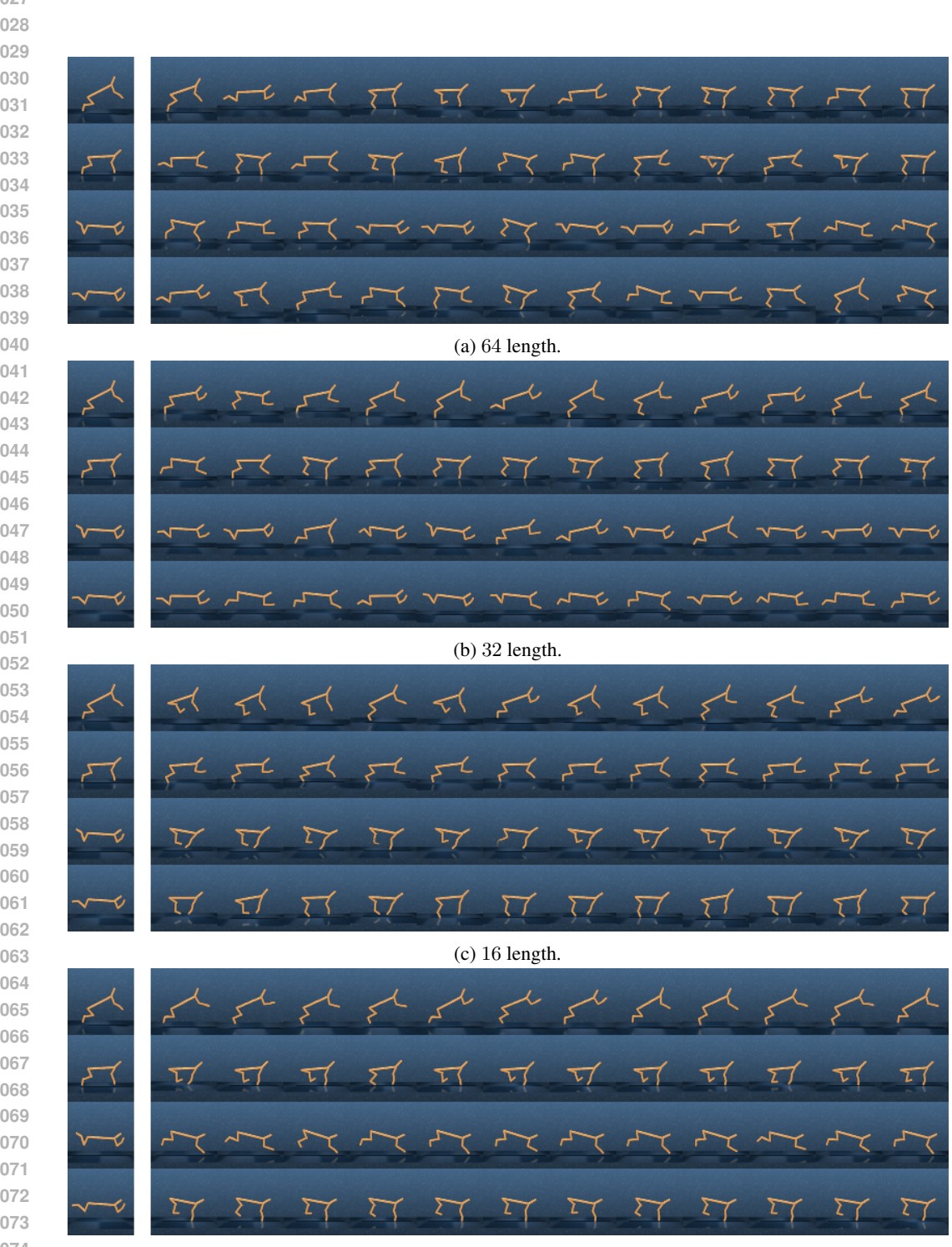

(a) 64 length.

(b) 32 length.

(c) 16 length.

(d) 8 length.

Figure 12: Sample goals from the `cheetah_run` task.

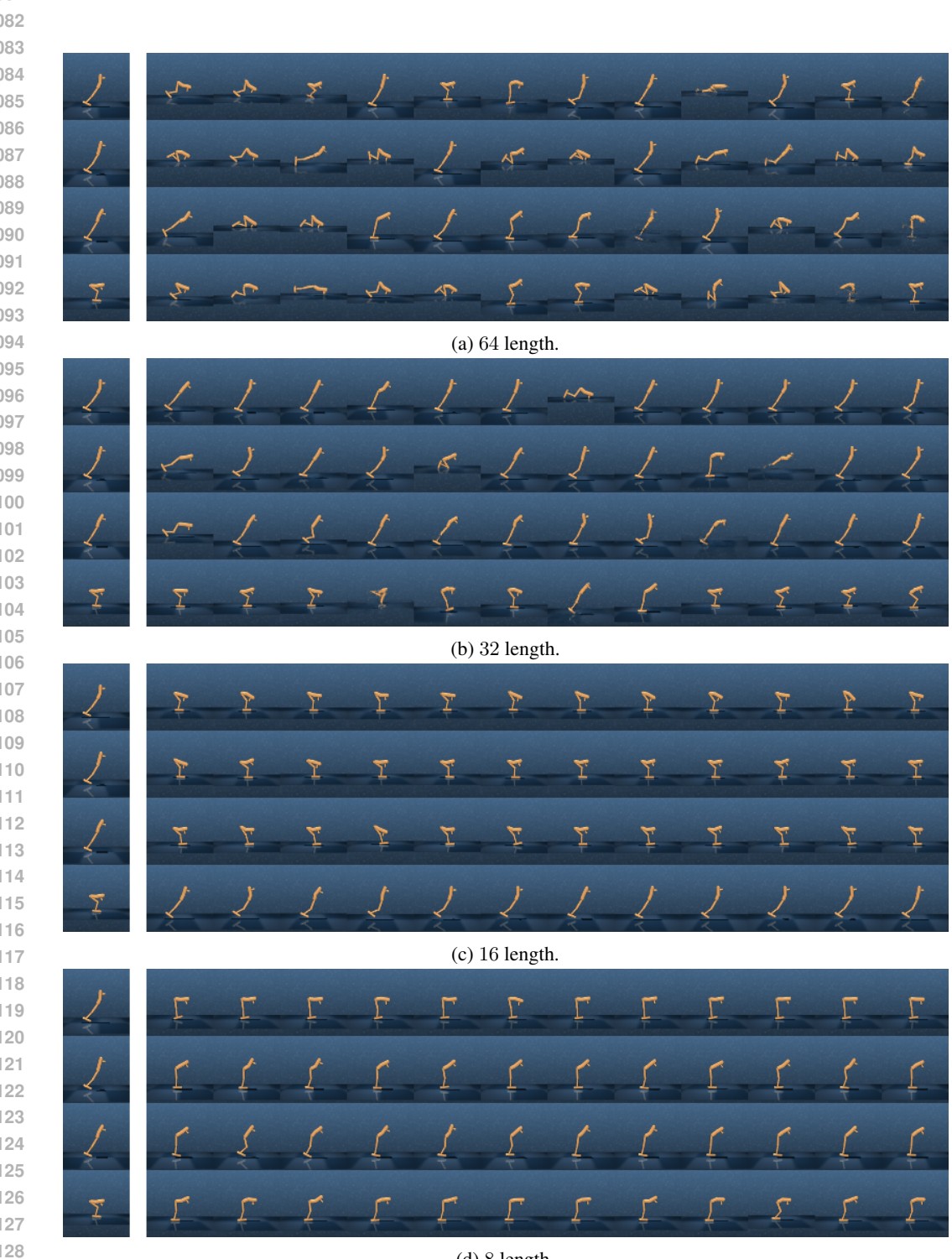

(a) 64 length.

(b) 32 length.

(c) 16 length.

(d) 8 length.

Figure 13: Sample goals from the `hopper_hop` task.

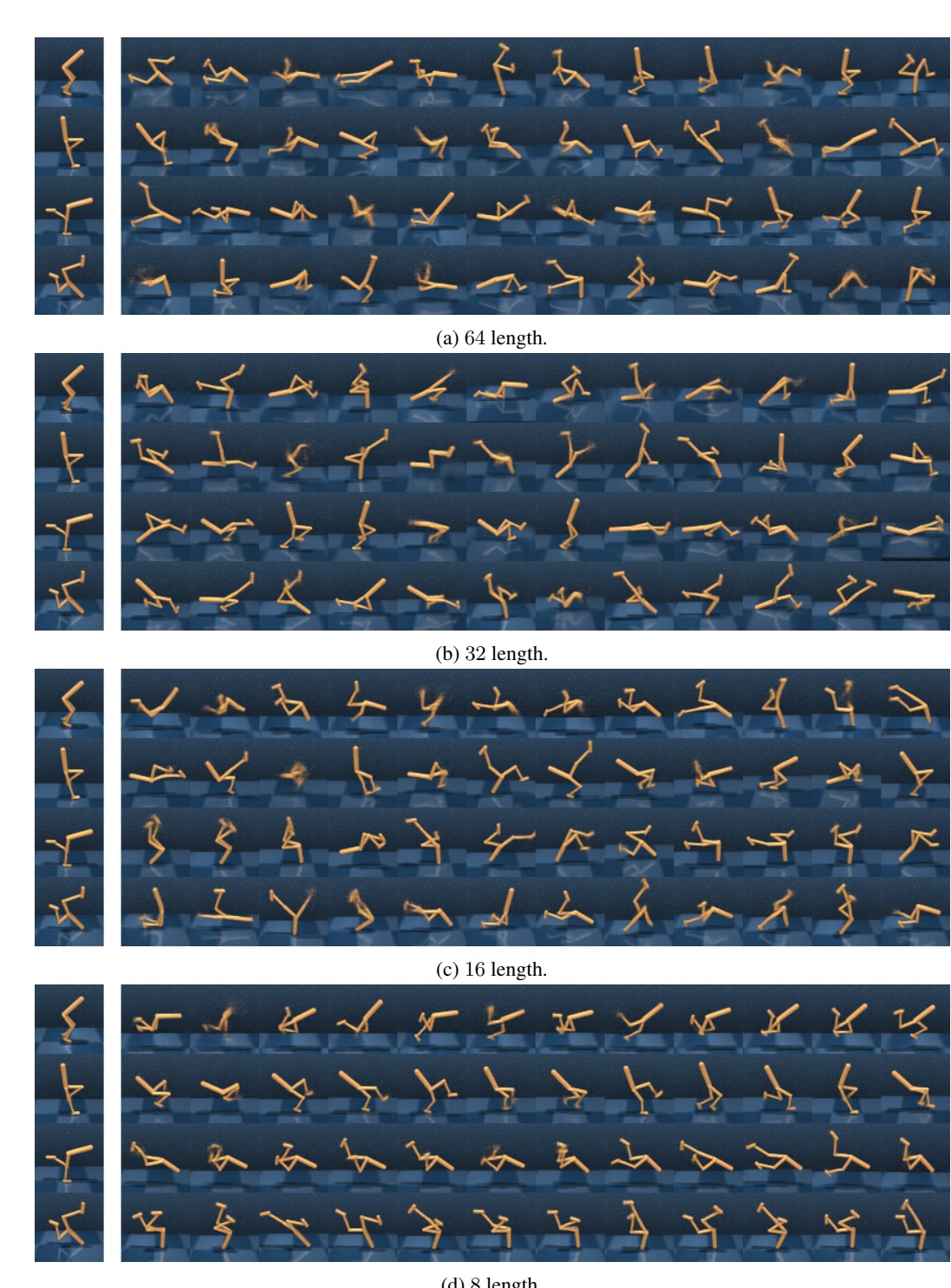

(a) 64 length.

(b) 32 length.

(c) 16 length.

(d) 8 length.

Figure 14: Sample goals learned using only the exploratory objective in a `walker` embodiment. The images on the left show the current state $s_t$, and the remaining images show the goal options generated by different skill CVAEs.

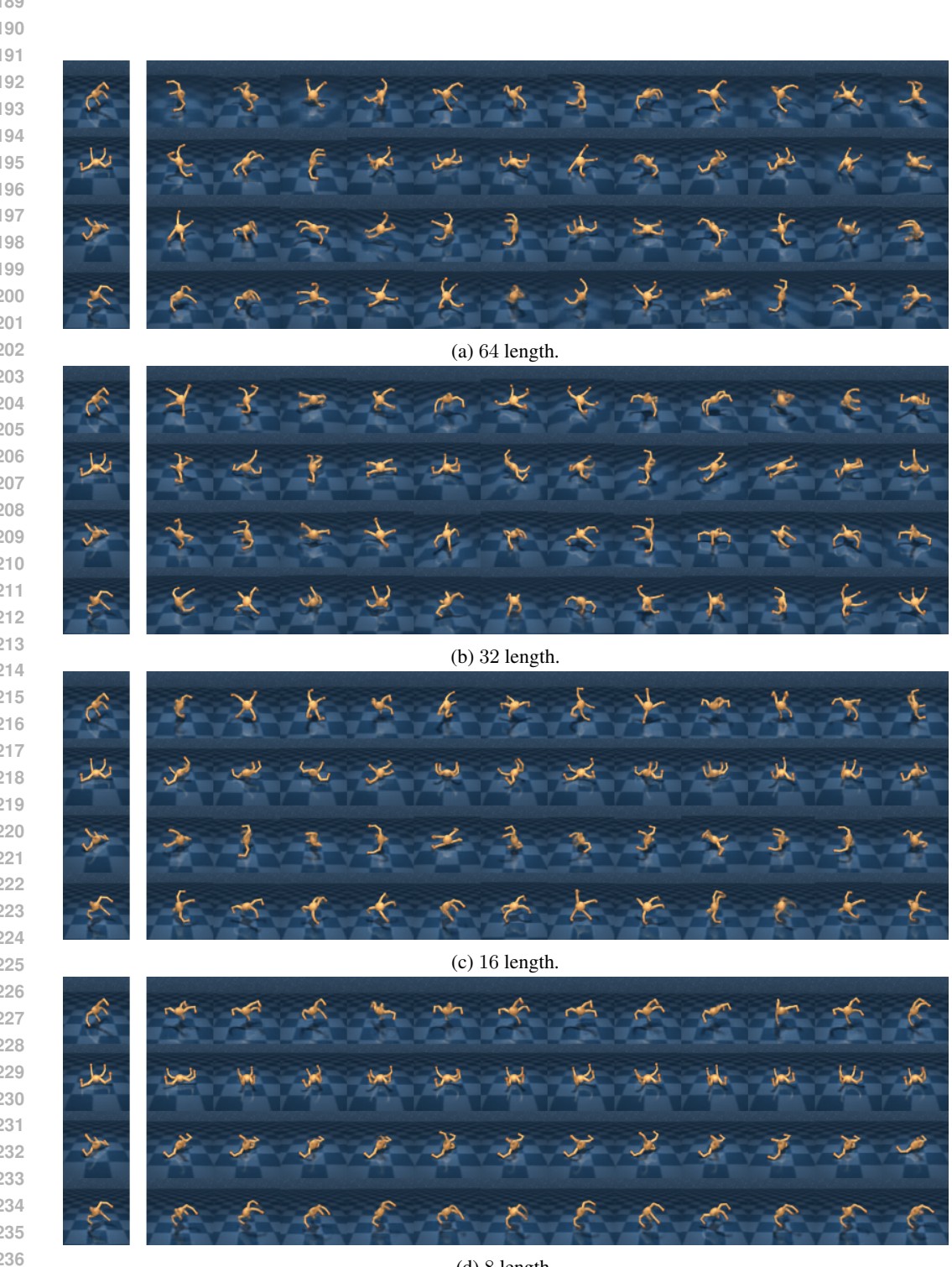

(a) 64 length.

(b) 32 length.

(c) 16 length.

(d) 8 length.

Figure 15: Sample goals from exploration as a `quadruped`.

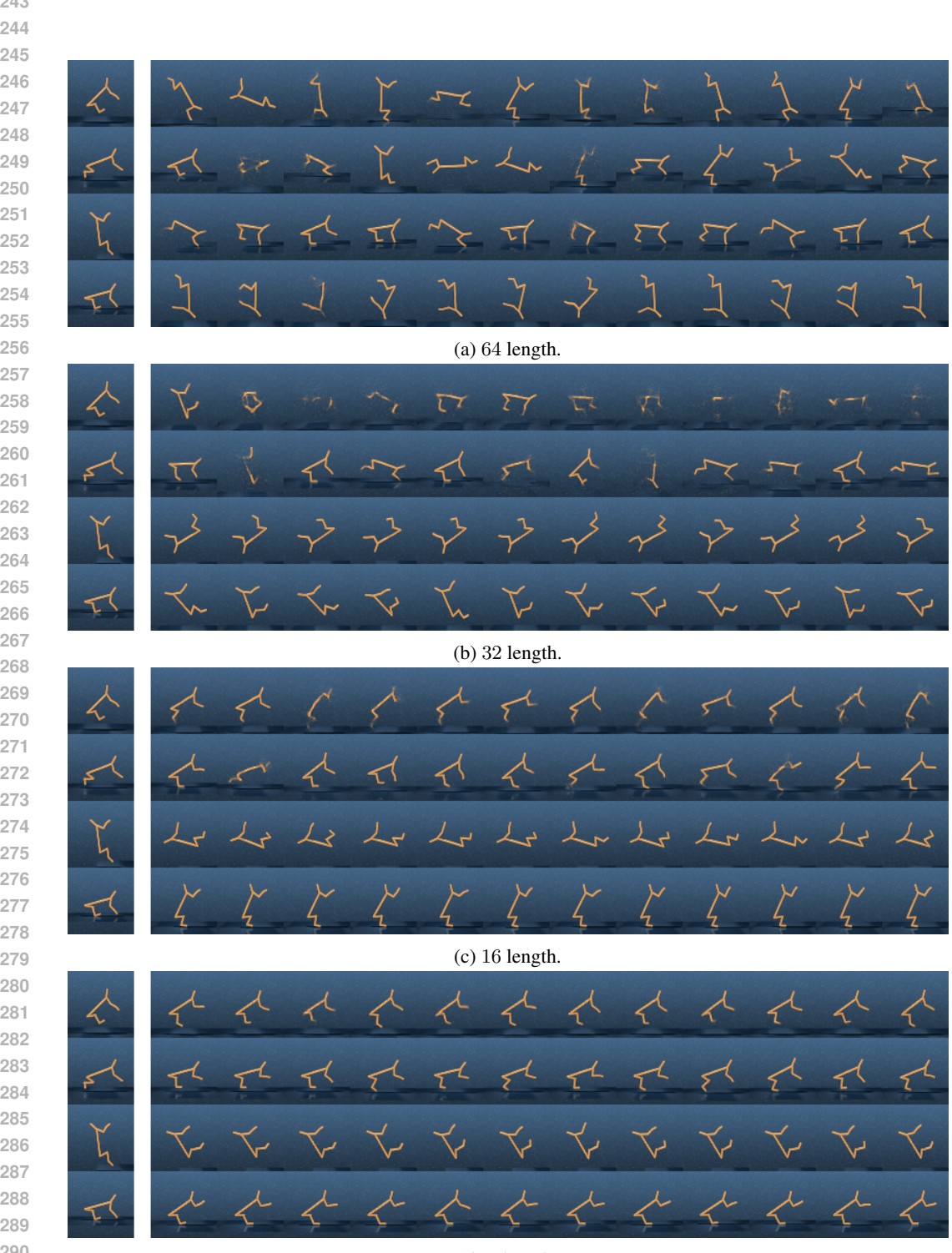

(a) 64 length.

(b) 32 length.

(c) 16 length.

(d) 8 length.

Figure 16: Sample goals from exploration as a `cheetah`.

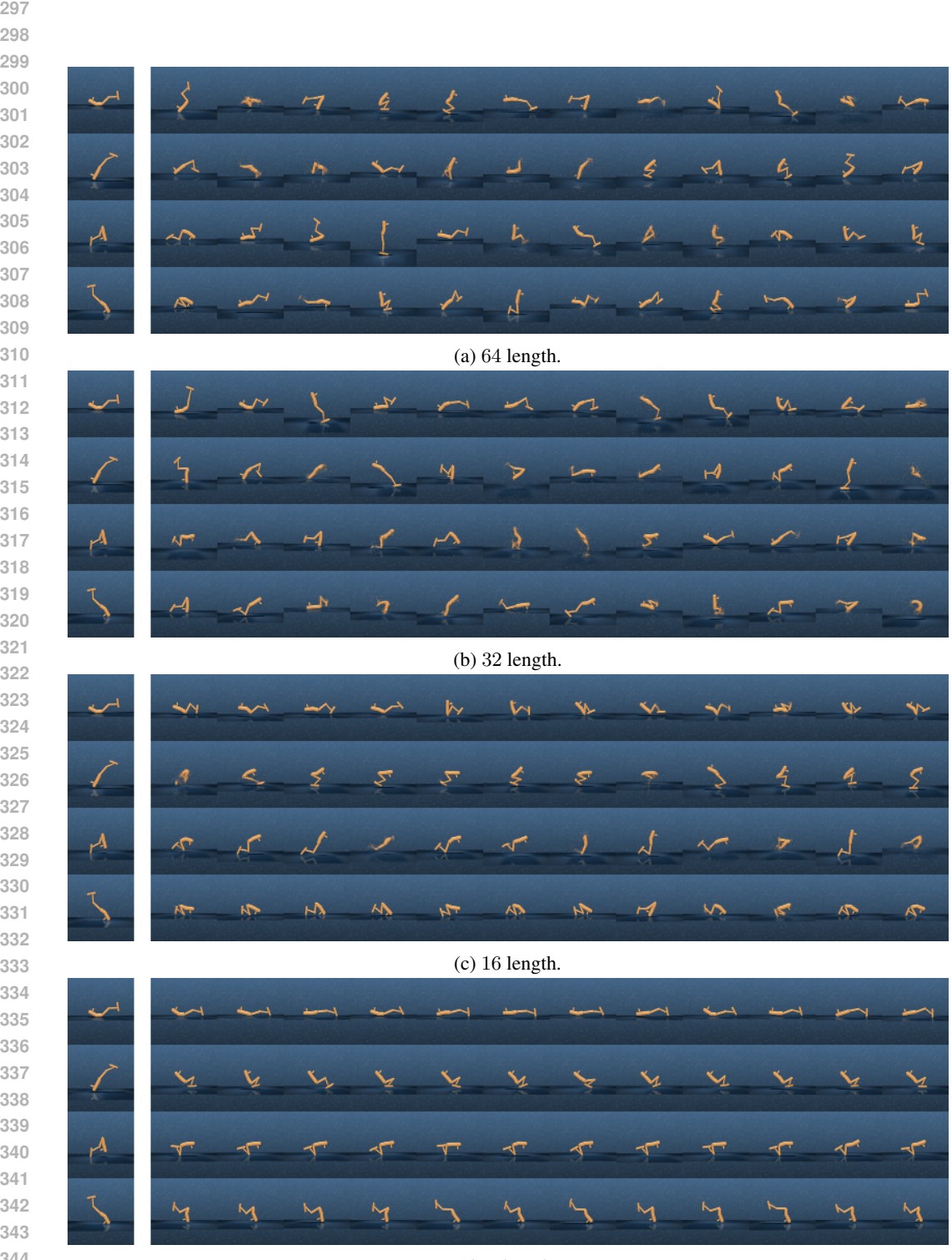

(a) 64 length.

(b) 32 length.

(c) 16 length.

(d) 8 length.

Figure 17: Sample goals from exploration as a `Hopper`.

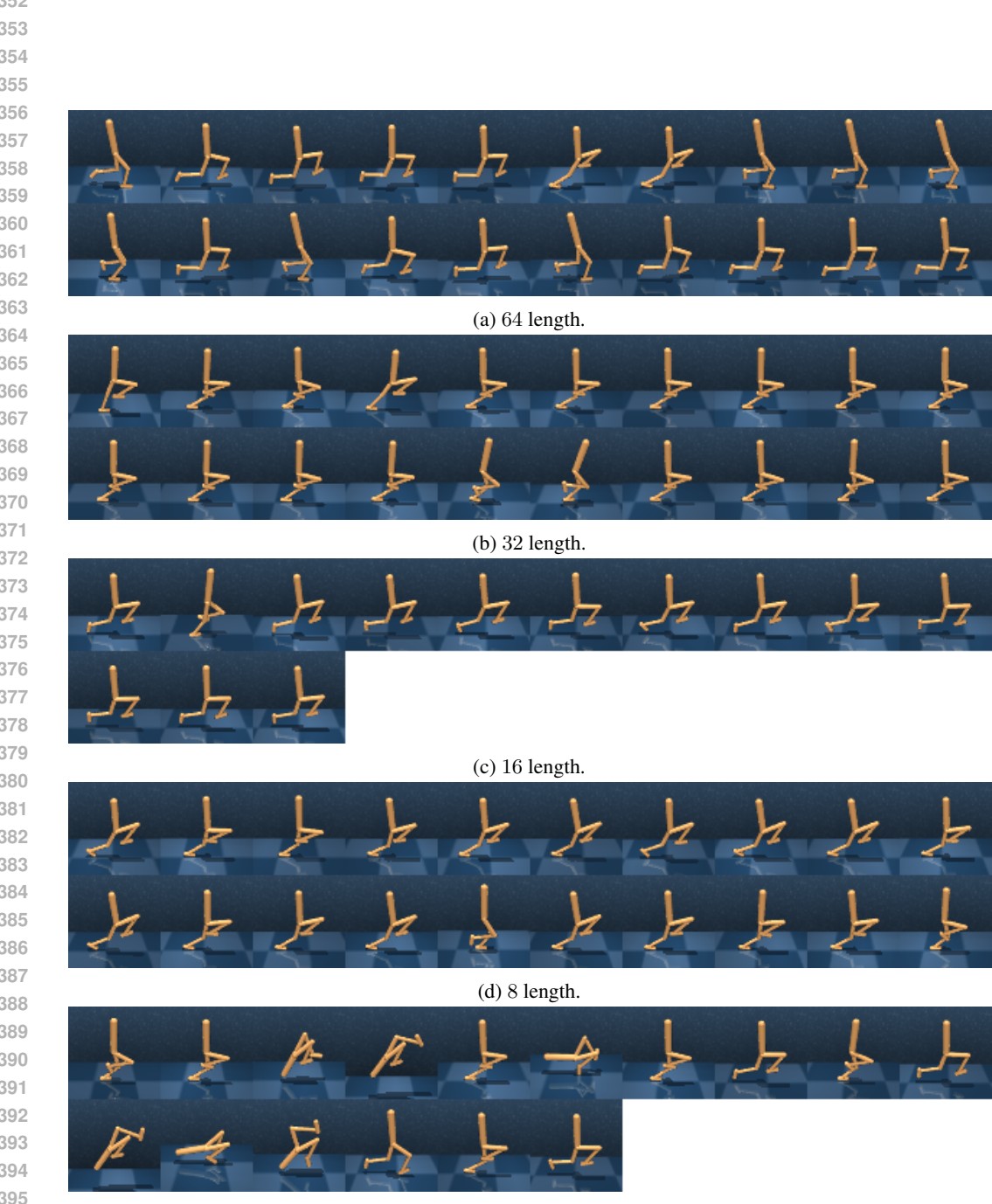

(a) 64 length.

(b) 32 length.

(c) 16 length.

(d) 8 length.

(e) ∞ length.

Figure 18: States segregated by choice for the `walker_run` task.

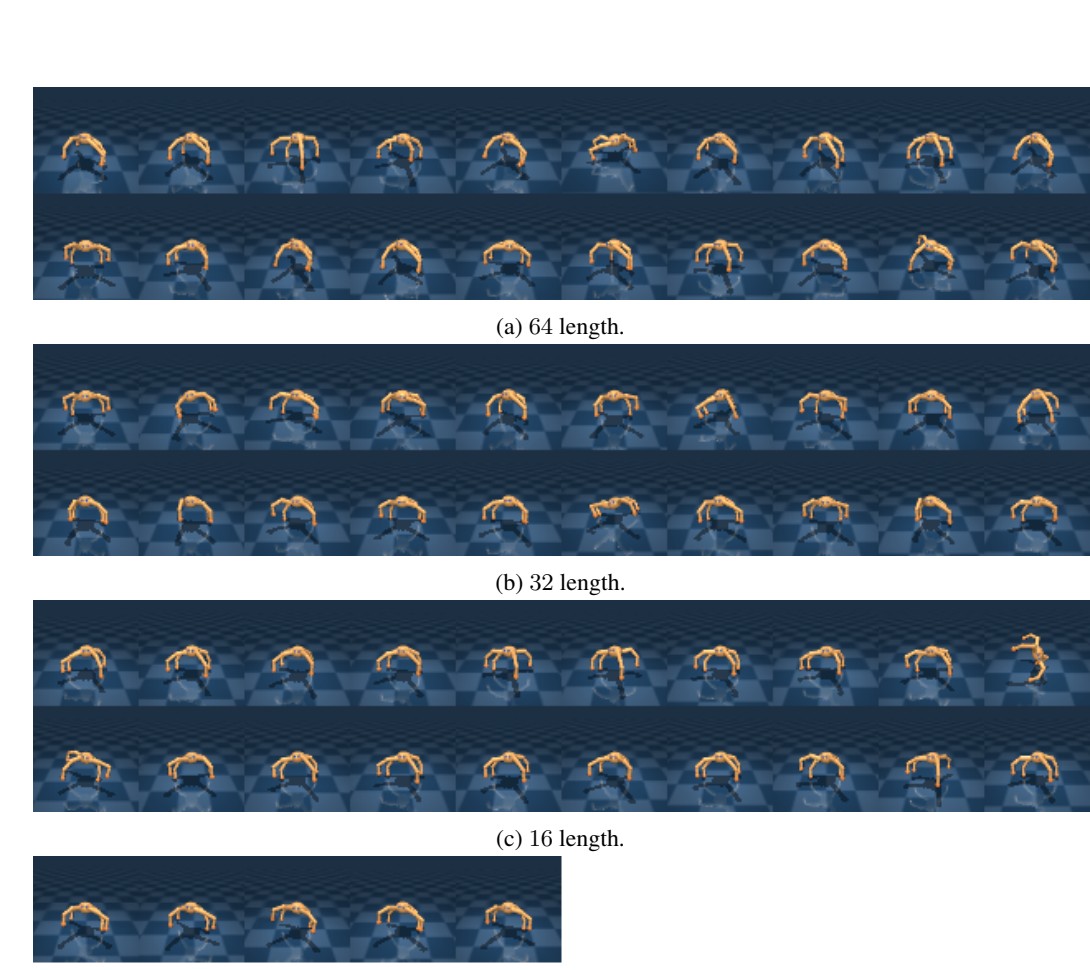

(a) 64 length.

(b) 32 length.

(c) 16 length.

(d) 8 length.

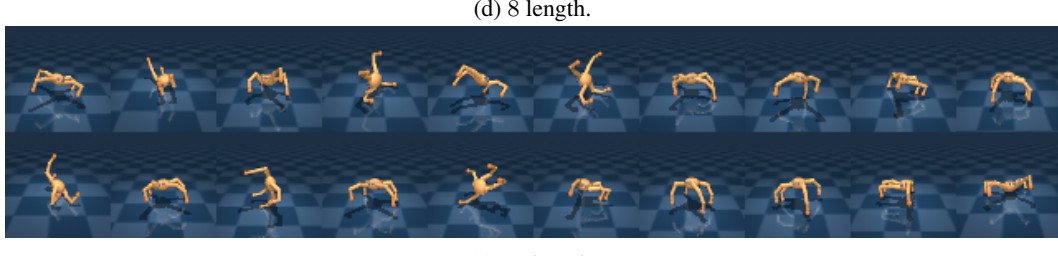

(e) ∞ length.

Figure 19: States segregated by choice for the `quadruped_run` task.

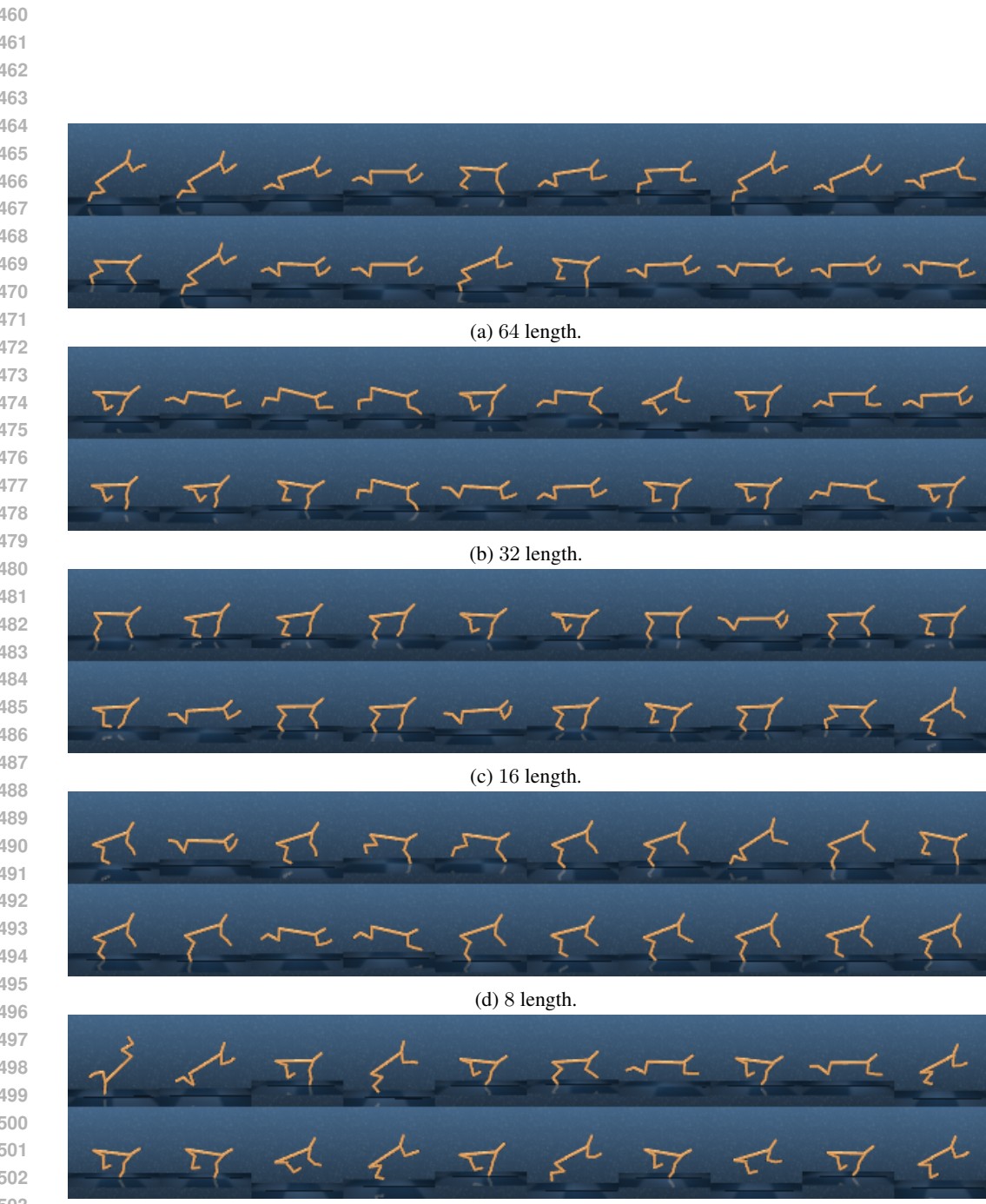

(a) 64 length.

(b) 32 length.

(c) 16 length.

(d) 8 length.

(e) ∞ length.

Figure 20: States segregated by choice for the `cheetah_run` task.

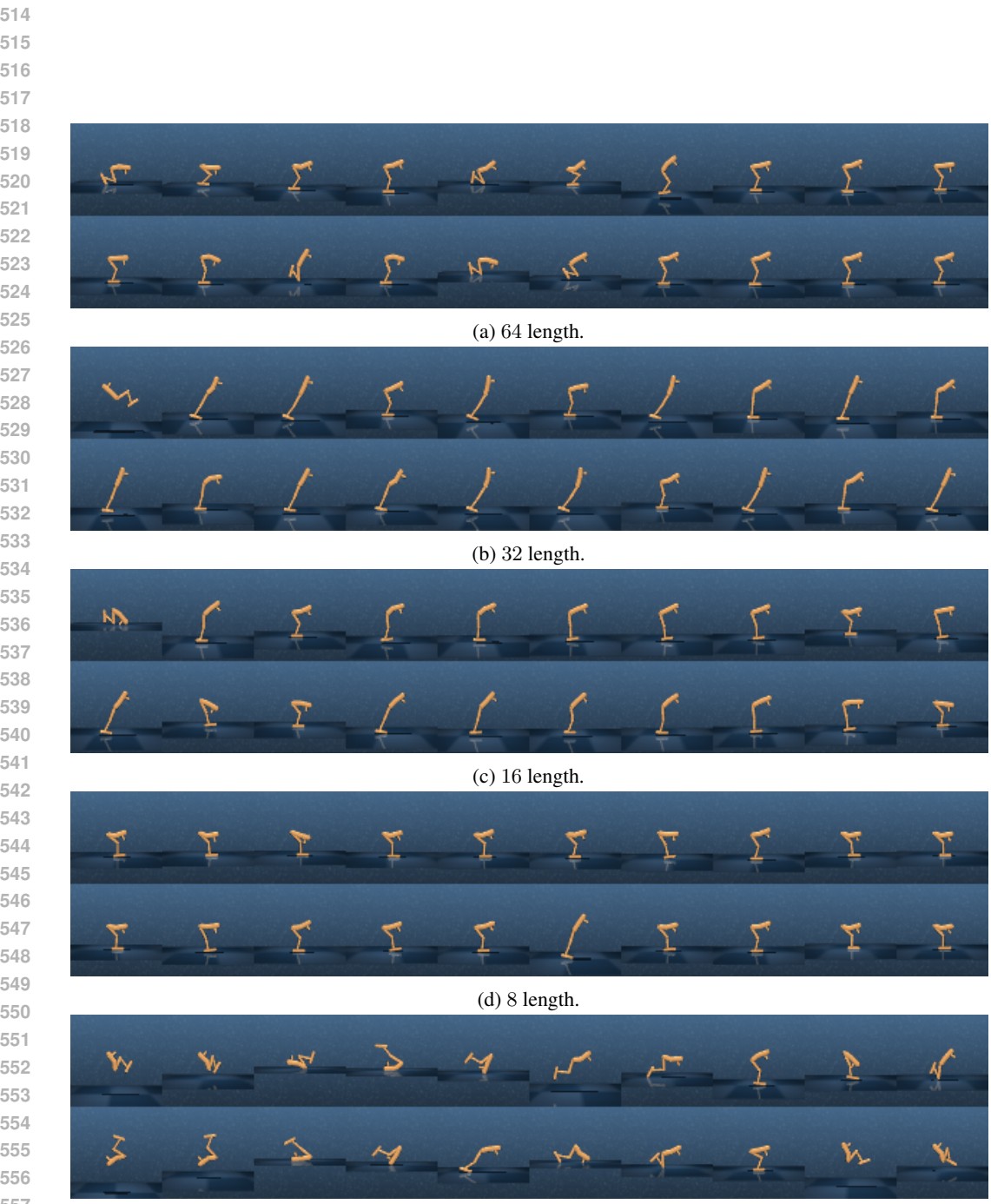

(a) 64 length.

(b) 32 length.

(c) 16 length.

(d) 8 length.

(e) ∞ length.

Figure 21: States segregated by choice for the `hopper_hop` task.

