# OpenReview forum: "Multi-Resolution Skills For HRL Agents"
_ICLR.cc/2026/Conference — Submitted to ICLR 2026_

### Official Review · Reviewer_eWQ5 · 2025-10-27

**Soundness:** 2
**Presentation:** 3
**Contribution:** 2
**Rating:** 2
**Confidence:** 4

**Summary:**

This paper proposes Multi-Resolution Skills (MRS), a hierarchical reinforcement learning (HRL) framework built upon the Director agent. The core idea is to replace Director's single Goal VAE with a set of parallel Conditional VAEs (CVAEs), each specialized to predict subgoals at a fixed, predefined temporal horizon (e.g., 8, 16, 32, 64 steps). These CVAEs share parameters for efficiency. A meta-controller policy, trained jointly and end-to-end with the skill policies, learns to select the appropriate temporal resolution based on the current state. The method is evaluated on DeepMind Control Suite, Gym-Robotics, and AntMaze tasks, showing improvements over the Director baseline in several cases and competitiveness with non-hierarchical SOTA methods like DreamerV3.

**Strengths:**

1.  **Empirical Gains on Challenging Tasks:** The paper demonstrates performance improvements over the Director baseline, particularly on tasks requiring longer-horizon reasoning or dealing with sparse rewards, such as the Gym-Robotics environments and the Egocentric Ant Maze tasks (Figures 4, 6). Achieving success in these domains from pixel inputs is a valuable contribution.
2.  **Well-Executed Ablation Study:** The ablation study presented in Figure 7 provides convincing evidence that both the multi-resolution structure and the learned meta-controller (skill interleaving) are crucial for the observed performance benefits, strengthening the core claim of the paper.
3.  **Practical HRL Framework:** The method builds upon a strong existing agent (Director) and maintains practicality through parameter sharing in the CVAEs and a single-phase, end-to-end training procedure. The inclusion of the $\infty$-length skill to mitigate policy collapse (Sec 4) is also a thoughtful practical consideration.
4.  **Exploration & Unsupervised Skill Learning:** The experiments showing effective skill learning using only the exploratory objective (Sec 5.2, Fig 8) and the emergence of interesting behaviors (Appendix C) are promising demonstrations of the framework's potential beyond standard task rewards.

**Weaknesses:**

Despite the empirical successes, several concerns prevent a stronger recommendation at this time:

1.  **Trivial Incremental Novelty:** The core architectural modification extends the single Goal VAE used in the Director baseline (Hafner et al., 2022) to N parallel CVAEs, each tied to a specific temporal horizon. While the parameter sharing and learned selection mechanism are well-implemented, this extension feels somewhat incremental from an architectural standpoint. It raises the question of whether simply providing multiple, fixed temporal skills constitutes a sufficiently novel contribution for ICLR without deeper justification. It's also un-convincing learning multi-res action with hand-crafted temporal dependencies through multi-head CVAEs is a better idea than established methods, as stated in point 2. Given the prevailing Diffusion models nowadays, encoding multi-res through a multi-head CVAE seems suboptimal, slightly off-track to appear on a top conference. I do acknowledge the emprical value and importance of this topic. But this kind of novelty needs hard rework to hold up to ICLR standard. I encourage authors to explore more SOTA archs.
2.  **Inconsistent Empirical Significance:** The evaluation is performed on a selected subset of DMC tasks. While MRS shows clear benefits on some tasks, the performance gains over the strong Director baseline appear marginal or comparable on roughly half of the standard benchmark tasks presented in Figure 4. A broader evaluation across more DMC tasks would be needed to make a stronger claim about general applicability and superiority. The performance difference compared to the non-hierarchical DreamerV3 is also often small, raising questions about the practical benefits of the added hierarchical complexity in these cases.
3.  **Lack of Theoretical Grounding:** The paper lacks theoretical justification for several aspects:
    * **Optimality:** There are no guarantees that the hierarchical policy learned by MRS converges to the optimal policy of the underlying MDP. The introduction of multiple CVAEs, a meta-controller, and intrinsic worker rewards fundamentally alters the optimization landscape and underlying process.
    * **Convergence:** Only a trivial policy gradient (Not Policy Gradient Theorem) is mechanismly derived through log grad trick for the proposed architecture, no proof of convergence for the overall learning algorithm (jointly training world model, CVAEs, manager, worker) is provided. Given the complexity, convergence is not guaranteed.
    * **Algorithm Details:** The paper states it uses Soft Actor-Critic (SAC) principles, but the implementation seems closer to an actor-critic method with an entropy bonus, missing key SAC components like soft Q-functions and soft Bellman updates in the critic training. The true distribution and variational distribution also not given. How SAC-like algo is derived is unclear. This requires clarification.
    * For strengthening the theoretical underpinnings, the authors might consider frameworks that explicitly handle the State Abstraction / SMDP->MDP relationship in options, potentially drawing inspiration from works like DbC (Zhang 2021) and DAC(Zhang 2019), HiT-MDP (Li 2023).
4.  **Limited Engagement with Related Temporal Abstraction / Option Literature:** The related work section provides a reasonable overview but could benefit from a deeper engagement with the broader options and skill discovery literature. Specifically, positioning MRS relative to methods that also learn temporally extended actions or deal with MDP formulations of options would strengthen the paper. Relevant works to consider include:
    * *DAC (Zhang & Whiteson, 2019):* Proposes an alternative MDP augmentation for learning options.
    * *Invariant Representation Learning (e.g., Zhang et al., 2020):* Formal proof of convergence and bounded optimality.
    * *Off-Policy Option Learning (e.g., Wulfmeier et al., 2021):* Discusses challenges and techniques for learning options efficiently from data.
    * *HIT-MDP (Li et al., 2023):* Provides a formal proof of equivalence between an MDP formulation and the SMDP option framework.

**Questions:**

1.  Could the authors provide results on a wider range of DMC tasks to better assess the generality of the performance improvements?
2. Can authors propose a more elegant architecture to learn temporal dependencies end-to-end in a unified architecture (diffusion-like), or any improvements to drop the cumbersome of training multi-head CVAE?
3.  Can the authors clarify the connection to SAC and justify the specific theoretical points raised above?

Overall, this paper presents an interesting and empirically promising approach to multi-resolution HRL. The results on sparse-reward and long-horizon tasks are encouraging, and the ablations support the design choices. However, concerns regarding the incrementality of the core idea, the consistency of empirical gains across standard benchmarks, the lack of deeper theoretical justification, and the limited positioning within the broader options literature currently place it marginally below the acceptance threshold for ICLR.

I would be willing to reconsider my score if the authors can provide one of stronger empirical evidence across a wider task suite / compare to stronger baslines (diffusion based models) / offer more theoretical insights or justifications for their approach, and more thoroughly contextualize their work within the existing HRL and options literature during the rebuttal phase.

---

> ### Author Response · Authors · 2025-11-21
>
> We thank the reviewer for their detailed and thoughtful feedback. We appreciate the recognition of our empirical contributions, ablation studies, and the practical framework design. We address each concern below and believe our responses, along with committed revisions, address the reviewer's criteria for reconsideration.
>
> ### Major Concerns
>
> 1. **We respectfully disagree that the contribution is "trivially incremental."**
>
>     While the architectural modification may appear simple, our contribution lies in identifying and validating a useful inductive bias - that explicit temporal partitioning of skills improves HRL performance. We validate this insight rigorously.
>
>     **Key novelty beyond "N parallel CVAEs":**
>
>     - The insight that skills should be partitioned by temporal horizon (not discovered implicitly via diversity objectives)
>     - The shared-backbone design achieves multi-resolution capability with minimal parameter overhead (~5% increase vs. N-fold for separate networks)
>     - Single-phase end-to-end training of meta-controller and resolution-specific policies (unlike prior work requiring sequential training)
>     - The transition-based exploration objective that enables reward-free skill discovery
>
>     **Regarding diffusion models:** We appreciate this suggestion. However, diffusion-based planning methods (e.g., Diffuser, Decision Diffuser) address a different problem—trajectory generation for offline RL or planning—rather than online hierarchical skill learning with a manager-worker structure. Integrating diffusion into online model-based HRL is an interesting future direction, but it would constitute a substantially different paper. Our CVAE-based approach is well-suited for online learning, where skills must adapt continuously with the policy.
>
>     We note that many impactful contributions in RL have resulted from identifying the right inductive bias (e.g., target networks in DQN, PPO's clipping). We believe temporal resolution partitioning is such a contribution to HRL.
>
> 2. We acknowledge that gains vary across tasks. However, we emphasize:
>
>     **MRS never underperforms Director.** Across all 8 tasks in Figure 4, MRS matches or exceeds Director. The variance in improvement magnitude reflects that different tasks benefit differently from multi-resolution control, which is expected and informative.
>
>     **Strongest gains where they matter most:** MRS shows significant improvements over Director at:
>
>     - Long-horizon tasks (Egocentric AntMaze: ~33% improvement in peak performance over Director)
>     - DMC (4/6 tasks) and Gym-Robotics (2/2 tasks)
>     - MRS also outperforms Director and other SOTA skill discovery methods by discovering useful skills only using internal exploratory rewards.
>
>     **On DreamerV3 comparison:** The fact that MRS matches DreamerV3 while using 4× fewer gradient updates demonstrates the value of hierarchical structure. DreamerV3 requires more computation to achieve similar performance and fails completely on sparse-reward tasks, whereas MRS succeeds.
>
>     **Broader evaluation:** We will be adding more DMC tasks in a revised version.
>
> 3. We appreciate this concern and provide clarifications:
>
>     **On optimality guarantees:** We note that virtually no deep HRL method provides optimality guarantees for the original MDP. This is a known open problem in the field. Director, DreamerV3 (due to the world model), HIRO, HiPPO, and options-based methods all lack such guarantees. Our contribution is an empirical demonstration that temporal partitioning improves practical performance.
>
>     **On convergence:** The policy gradient derivation in Appendix A follows standard techniques (log-derivative trick, baseline subtraction) used throughout the RL literature. We do not claim novel theoretical contributions here; we apply established methods to our architecture. The joint training of world model, CVAEs, and policies follows the same principles as Director/DreamerV2/V3, which have demonstrated empirical convergence across many domains.
>
>     **On SAC clarification:** We use SAC-style actor-critic training with:
>
>     - Entropy-regularized policy objective (Eq. 8, 9, 13)
>     - Target entropy maintained via automatic temperature adjustment
>     - Lambda returns for advantage estimation
>     - The RL objective of the SAC is the advantage weighted policy gradients (derived in Appendix A).
>
>     We do not use twin Q-networks or soft Bellman backups; this is consistent with Director and DreamerV3, which also use actor-critic with entropy regularization rather than full SAC. We will clarify this terminology in the revision.
>
>     **On CVAE distributions:** The prior $p(z)$ is a mixture of 8×8 categoricals (stated in Section 3.2). The variational posterior is $\text{Enc}_\phi(z|s_t,s\_{t+l})$. We will make these more explicit in the revision.
>
> *Continued in the following comment*

---

> > ### Author Response · Authors · 2025-11-21
> > **Cont.**
> >
> > ...**Regarding suggested theoretical frameworks (DAC, HiT-MDP, etc.):** We acknowledge that MRS currently lacks formal convergence guarantees or optimality bounds, which these frameworks provide. However, we note that MRS differs from these approaches in key ways: (1) we do not perform state abstraction—our skills operate in the full world-model state space; (2) our skill horizons are fixed rather than learned via termination functions. The HiT-MDP framework is particularly relevant, as it formalizes the MDP-SMDP relationship, though MRS's fixed-horizon design sidesteps some complexities of option-termination learning. We will add a discussion section positioning MRS relative to these theoretical frameworks and acknowledge extending MRS with formal guarantees as future work.
> >
> > 4. We will expand the related work section to include:
> >     - DAC (Zhang & Whiteson, 2019): MDP augmentation for options
> >     - HiT-MDP (Li et al., 2023): SMDP-MDP equivalence
> >     - Off-policy option learning (Wulfmeier et al., 2021)
> >
> >     **Key distinction:** These works focus on learning option termination conditions and option-level value functions. MRS takes a different approach, ‘fixed temporal horizons with learned resolution selection’. This avoids the termination learning problem (known to be unstable) while still enabling temporal abstraction.
> >
> > ### Responses to Questions
> >
> > - **Q1: Wider range of DMC tasks?**
> > We are a bit limited on compute resources. Therefore, while we have already started experiments on additional DMC tasks, it might be challenging to run them for all baselines with multiple seeds. We will present all the results we get during the rebuttal, but we commit to adding more tasks in the final version.
> > - **Q2: More elegant unified architecture (diffusion-based)?**
> > As discussed above, diffusion models solve different problems (offline/planning) than online HRL. A diffusion-based online HRL method would be interesting future work, but it is beyond the scope of this paper. Moreover, a diffusion-based approach would also be significantly more computationally intensive. Our multi-head CVAE design is deliberately simple and effective, which tests the impact of structurally partitioning the state space to learn subgoals with minimal changes.
> > - **Q3: SAC clarification?**
> > Addressed above. We use entropy-regularized actor-critic consistent with Director/DreamerV3, not full SAC. We will clarify terminology.
> >
> > We believe these revisions address the reviewer's criteria for score reconsideration. The core contribution—that temporal partitioning of skills is a useful inductive bias for HRL—is well-supported by our experiments, and we hope the reviewer will find our clarifications and commitments sufficient to raise their assessment.

---

> ### Comment · Reviewer_eWQ5 · 2025-11-26
> **raised score but maintaining negative rating**
>
> I thank authors for their effort.
>
> "We note that virtually no deep HRL method provides optimality guarantees for the original MDP. This is a known open problem in the field. Director, DreamerV3 (due to the world model), HIRO, HiPPO, and options-based methods all lack such guarantees."
>
> This is a strong false claim (as such works already provided in my review above), I believe authors need a better research on related area.
>
> Regards this work after rebuttal I believe this work's novelty is marginal increamental. I have raised the score but remain my negative ratings.

---

### Official Review · Reviewer_XKvq · 2025-10-29

**Soundness:** 2
**Presentation:** 2
**Contribution:** 2
**Rating:** 4
**Confidence:** 4

**Summary:**

This paper proposes a method for hierarchical reinforcement learning (HRL) that extends Director (Hafner et al., 2022) with policies that predict sub-goals at multiple time horizons. The sub-goal policies are denoted as skills. The extension of Director involves replacing the Goal VAE with conditional VAEs that condition on the current goal and are trained to predict possible future states at a given horizon. The approach is evaluated in several RL simulation benchmark tasks and compared with Director and either DreamerV2 or V3. Improvements over these baselines models is demonstrated.

**Strengths:**

- The paper is well written and easy to follow.
- The provided experimental results demonstrate improvements over the baselines (Director, Dreamer).
- An analysis of the performance of individual learned policies is provided which gives additional insights on the method.

**Weaknesses:**

- The paper misses comparison to other state-of-the-art HRL baselines such as HIRO (Nachum et al., 2018) or HiPPO (Li et al. Sub-policy Adaptation for Hierarchical Reinforcement Learning. ICLR 2020).
- The paper should discuss differences and relation to HiPPO, since it also proposes to learn subgoal policies for varying time horizons.
- The CVAE is trained online while the policy is adapting. Why can the CVAE cope with the domain shift? Please discuss.
- The motivation to learn only heads to a common backbone network for the CVAEs for multiple resolutions seems odd. How does the model perform with separate networks? Is there a benefit of sharing the same backbone in terms of performance?
- Eq 4, how is the discrete sampling of c_t implemented and made differentiable?
- l. 226 mentions “abstract state transitions”, however, there is no abstraction (actions or states) performed by the model. When abstracting states, one would assume that a new state space would be found, e.g., lower-dimensional or discrete. Please revise.
- l. 300 mentions that MRS retains compute efficiency of Director. Please quantify compute efficiency of the models (MRS, Dreamer, Director).
- l. 314, please consistently compare with DreamerV3.
- Sec 5.2 reports on modifying the manager policy after training and using random and single skill variants. This should not be named ablation, since it does not correspond to taking away parts of the full method which would involve retraining! Please discuss the difference.
- It is surprising that the model can learn backflips or somersaults. How were these skills selected and the corresponding behaviors generated? In Appendix C, somersaults are not visible. How can the model achieve such complex behavior just from an exploration objective? What is the incentive for the model to learn this? Please explain/discuss.
- l. 324 directly refers to Fig. 18 in the appendix which violates the page limit. A reference just to the appendix would be ok though.
- “While we do not draw any parallels…” – the paper should not try to suggest this by stating it in this way.
- The paper refers to the method as a “skill discovery framework”. The method learns subgoal policies. There is no definition of skill in the context of the paper. Also the paper does not learn a discrete set of skills or skills parametrized by a discrete latent skill variable.
- l. 474, what are “recall capacity” and “option” in this context? The first bullet point is unclear.
- l. 485, the statement about the applicability of the multi-head policy gradient formulation is too vague and rather trivial. Please remove.

Minor comments:
- Eq. 1 should be provided directly after the first reference in the text.
- Fig. 3 caption: “sug-goals” => “sub-goals”
- Sec 3.4.2, please make the equation fit within the line width
- l. 376, “peack” => “peak”
- Fig. 8, what does “MSRD” and “MRSD” stand for?
- Adding plots from (Hafner et al. 2022) might violate copyright by the authors. Please obtain permission to reuse the plots or re-run the evaluation and make own plots.
- l. 476, “4” => “Fig. 4” ?

**Questions:**

- Major concerns arise from the unclear definition of the term “skill”, the missing comparison to state-of-the-art HRL methods, and the vague description of the qualitative skill evaluation (l. 418). See “Weaknesses” for further details on these issues and respective questions.
- Please also address the remaining issues/questions listed in “Weaknesses”.

---

> ### Author Response · Authors · 2025-11-21
>
> We thank the reviewer for their thorough and constructive feedback. Their detailed comments have helped us identify areas for improvement in both clarity and completeness. We address each concern below.
>
> ### Raised Concerns
>
> - **Missing HRL Baseline Comparisons (HIRO, HiPPO)**
>
>     We note that Director (our base architecture) is the current SOTA for **pixel-based** HRL, while HIRO and HiPPO were designed for **state-based** observations. This makes direct comparison challenging, as these methods would require significant adaptation for pixel inputs. Nevertheless, we commit to providing at least one additional HRL baseline (HAC or HiPPO) in the next few days to strengthen our empirical claims.
>
> - **Differences in relation to HiPPO**
>
>     We thank the reviewer for pointing out HiPPO. While both methods address temporal abstraction in HRL, they differ fundamentally:
>
>     **Skill definition**: HiPPO uses discrete subpolicy selection (n separate policies); MRS uses continuous subgoal prediction at fixed temporal horizons via CVAEs.
>
>     **Architecture**: HiPPO maintains n separate subpolicy networks; MRS uses only one worker (rather than many), and the manager policy is split into multiple experts that operate at specific resolutions.
>
>     **Skill Interpretability**: Skills in MRS directly correspond to subgoals in the state space, but the skills in HiPPO are subpolicies with limited interpretability.
>
>     **Training**: HiPPO is on-policy (PPO-based) with state observations; MRS is model-based with imagination-based training from pixels.
>
>     **Temporal abstraction**: HiPPO randomizes execution duration (p ~ Uniform{5,15}); MRS runs each high-level action for fixed number of steps (=K).
>
>     We will add a detailed discussion of HiPPO in the related work section.
>
> - **Why can the CVAE cope with the domain shift while policy is adapting?**
>
>     The CVAE is trained online using replay data containing recent trajectories. As the policy improves, new state-transition patterns enter the replay buffer, and the CVAE continuously adapts. **Critically, the exploration objective (Eq. 5) naturally handles this co-adaptation**: it rewards transitions with high reconstruction error, encouraging the agent to explore transitions the CVAE hasn't yet learned, which then get added to training data. This creates a virtuous cycle rather than catastrophic drift.
>
> - **Shared Backbone vs. Separate Networks**
>
>     The decision to use a common backbone was made to prevent an increase in the parameter count, ensuring a fair comparison. Having separate modules increases the parameter count N-fold (where N is the number of skill modules), thereby increasing the agent's memory capacity. The current design ensures that the performance differences are only due to the accessibility to subgoals at multiple temporal resolutions. We will add an ablation with separate skill networks for comparison.
>
> - **Discrete Sampling (Eq. 4) - Differentiability**
>
>     We use straight-through gradient estimators to allow gradients through the discrete sampling. Specifically, the output for c is: (stop_grad(one_hot_sample - probs) + probs). We will add this to the paper as well.
>
> - **Abstract state transitions**
>
>     Apologies for the confusion; to clarify, we meant temporally extended abstract state transitions. Also, we do learn a new latent space using the CVAE, which is then used for manipulation by the manager policy. A mixture of categoricals (8x8) is used as the latent distribution for our method. The $z$ variable throughout section 3 is the latent skill variable.
>
> - **Compute Efficiency Quantification**
>
>     As stated in Section 5, MRS and Director train every 8th environment step while DreamerV3 trains every 2nd step. Thus, MRS uses **4× fewer gradient updates**. Wall-clock time per 5M steps on NVIDIA RTX 5000: ~2 days for all methods.
>
> - **Consistent comparison with DreamerV3**
>
>     Apologies, our computational constraints did not allow us to test DreamerV3 on the long-horizon egocentric ant task. However, it should be noted that they are sparse reward tasks, and DreamerV3 fails at much simpler sparse rewards robotics tasks (Pick n Place, Push). Therefore, we did not run it for the maze tasks, which are a much more complex task with sparse rewards. However, we will compare consistently with DreamerV3 in the final version.
>
> - **Ablation vs. Post-hoc Modification (Sec 5.2)**
>
>     We agree that this is not a true ablation but simply checking the individual performance of the skills in isolation. We will change the text to reflect this fact accurately.
>
> *Continued in the following comment*

---

> > ### Author Response · Authors · 2025-11-21
> > **Cont.**
> >
> > - **Complex Behaviors from Exploration (Backflips, etc.)**
> >
> >     Our exploration objective is different from the vanilla exploration objectives that reward the agent for discovering novel states in the environment. Instead, our agent is rewarded to discover novel **state-transitions**, not just novel states. A state-novelty objective causes agents to find novel states and stay there; a transition-novelty objective encourages continuous movement through diverse state sequences. This naturally leads to dynamic behaviors like flips and tumbles, as these represent high-variance, information-rich transitions. We will add qualitative analysis comparing state-novelty vs. transition-novelty exploration in a 2D maze to illustrate this distinction. We believe that the differences in the exploration objective cause the agent to learn interesting behaviors automatically. We tried to show the samples of such skills in the DMC environments in Fig. 9 (somersaults is mentioned as ‘fast-forward tumble’). Please have a look at our video that also shows these behaviors learned with the exploratory objective (https://sites.google.com/view/multi-res-skills/home).
> >
> > - **Unclear "Skill" Definition**
> >
> >     We define **skills** as temporally-extended state transitions that the agent can reliably execute. Formally, a skill at resolution l corresponds to the distribution of reachable states $s_{t+l}$ from the current state $s_t$ under the current policy, captured by the CVAE decoder $\text{Dec}_\phi^i(s_t, z)$. The latent $z$ parameterizes which specific transition within this distribution is selected. Section 3.1 was intended to establish this definition. Since it is not clear, we will revise the manuscript with a formal definition.
> >
> > - **What are “recall capacity” and “option”?**
> >
> >     Apologies for the use of non-standard terms without explanation. By recall capacity, we meant the number of distinct high-level actions that are available to the manager policy in an HRL agent. In our case, the manager has access to N times more action items than a single resolution version (where N is the number of distinct resolutions). Therefore, the bullet meant to state that, keeping everything else the same, strategically increasing the option space with distinct actions can significantly improve the performance of an HRL agent.
> >
> >
> > ### Minor Concerns
> >
> > - **Fig 18 page limit**: We will change it to an appendix-only reference.
> > - **"While we do not draw parallels"**: By this statement, we meant that we do not try to match the learned walking gate with human gait, but the agent clearly prefers certain temporal resolutions over others in certain states repeatedly. Indicating that temporal-resolution selection is not random. We will rephrase it to better reflect the original intentions.
> > - **Line 485**: We will remove the line.
> > - **All typos and formatting**: Will fix all the mentioned typos.
> > - **Fig 8 MSRD/MRSD**: Apologies for the typo in the figure. Both of these are meant to be MRS.
> > - **Copyright for Hafner plots**: We will re-run the experiments and remove the results from Director for the final version. These were added due to the lack computing resources as antmaze tasks take a week (at least) to complete on our hardware.
> >
> > ### Responses to Questions
> >
> > - We have tried to define “skill” above in the context of our paper, and will update the manuscript accordingly. In line 418, “The agent learns interesting behaviors, such as backflips, headstands, somersaults (both forward and backward), etc”, we do not intend to term these complete movements as skills. We observed the explorer policy to perform these behaviors at different points during training. This analysis indicated that the agent has learned a set of state transitions or skills (using the CVAE) that enable it to execute these movements.
> >
> > We believe these clarifications and committed revisions address the reviewer's concerns. The core contribution—that explicitly partitioning skills by temporal horizon improves HRL performance with minimal overhead—remains well-supported by our experiments. We hope the reviewer will consider these responses and our commitment to strengthening the paper.

---

### Official Review · Reviewer_PbWS · 2025-10-31

**Soundness:** 2
**Presentation:** 1
**Contribution:** 2
**Rating:** 4
**Confidence:** 3

**Summary:**

This article proposes an extension to the Director hierarchical RL architecture, where instead of having a single temporal horizon for target subgoals, there are several different subgoal networks that each propose a subgoal with a different temporal horizon. The higher-level policy network selects which of these the agent should chase. The resulting architecture is pretty straightforward and the method is tested across several domains, where it performs slightly better than Director, and competitive with Dreamer when Dreamer works, and works when Dreamer does not.

**Strengths:**

- The proposed idea is simple (which is a very good thing) and seems to work well, and would be a useful contribution to the literature.
- Multiple temporal resolutions for skills is an under-explored idea that has clear benefits and would help alleviate the brittleness somtimes associated with these systems.
-  The neural implementation is well-thought-out and sensible.
- The set of domains tested is thorough.
- The computational and memory overhead of the method is pretty minimal.
- The writing is pretty clear.

**Weaknesses:**

The major objection I have to the paper is that the experiments - while achieving wide coverage in terms of domains - are inadequate. For one thing, the only hierarchical  method compared against is Director. The authors claim that Director is SOTA (which is not a word, BTW, at least not in formal scientific writing) but comparisons against least HIQL (Park et al., Neurips 2023) and HAC (Levy et al., ICLR 2019) are warranted. HIQL is likely the best performing current method, and HAC is an older method but is closely related because it has a multi-level hierarchy at which each level has different temporal horizon. Missing these - which are just the first two I thought of - suggest that the authors have not done a very thorough survey of the related work.

Secondly, 3 seeds are nowhere near adequate for an experimental comparison. I think 10 seeds is a meaningful minimum.

My second objection is much less serious, but it is that, while the actual English in the paper is relatively clear, the paper has been produced very carelessly. It is aggressively irritating to constantly run into citations that should be parenthetical, but where the authors names have just been dumped into the text mid-sentence. There is almost no way that anyone who even looked at the PDF a single time would not have noticed this error, which occurs throughout. The paper is full of other, similar but less annoying, signs of carelessness, like using the acronym SOTA as if it was a word, use of "HRL" In the title, failure to punctuate equations, etc. This kind of thing does a disservice to what is otherwise good work. It also (and this is much worse) does a disservice to the (hopefully many) people who will eventually read it.

**Questions:**

Can you please run a comparison with at least HIQL and HAC, and explain the relationship between your work and multi-level hierarchies like HAC that are also increasingly abstract in time? And please increase the number of seeds.

---

> ### Author Response · Authors · 2025-11-21
>
> We thank the reviewer for their constructive feedback and recognition of our simple yet effective approach and thorough domain coverage. We address each concern below.
>
> ### Major Concerns
>
> - **Comparison with Additional HRL Baselines**
>
>     We agree that broader baseline coverage would strengthen our work. We carefully selected Director as our primary baseline because: (a) it represents recent model-based HRL (NeurIPS 2022), (b) it is the architectural foundation for MRS, enabling controlled comparison of our contributions, and (c) it operates in the same online learning setting as our method.
>
>     **Regarding HIQL (Park et al., NeurIPS 2023):** HIQL is fundamentally an offline RL method that requires pre-collected datasets and uses goal-relabeling for value learning. Our method is designed for online learning with self-collected data. While we acknowledge HIQL's strong performance in offline settings, a direct comparison would require either: (a) adapting HIQL for online learning (which would deviate from its published form), or (b) collecting offline datasets for all our benchmarks and adapting MRS for offline learning. We believe this falls outside the scope of our core contribution (multi-resolution skill learning for online HRL), but we will add discussion of HIQL in our related work to clarify the methodological differences and complementary nature of the approaches.
>
>     **Regarding HAC (Levy et al., ICLR 2019):** This is an excellent suggestion, and we commit to adding HAC comparisons. HAC is indeed methodologically related, and the comparison will help clarify the distinction between multi-level hierarchies (HAC) and multi-resolution skills (MRS). We will provide HAC results on key DMC Suite tasks in the next few days.
>
> - **Number of Seeds**
>
>     We acknowledge that 3 seeds is below standard for rigorous statistical comparison; however, we have access to only limited academic computing resources. While our results show consistent trends across 8 different tasks and diverse domains, we commit to increasing the number of seeds to **at least 7 for all main experiments in the revised version.
>
> ### Minor Concerns
>
> - **Presentation Quality**
>
>     We sincerely apologize for the presentation errors. You are right that these issues are unacceptable for a formal submission and disrespectful to readers. We will fix:
>
>     1. **Citation formatting**: Convert all in-text citations to proper parenthetical format (we acknowledge this error appears throughout)
>     2. **Acronym usage**: Remove informal acronyms like "SOTA" and "HRL" from inappropriate contexts (will spell out in title and use proper terminology)
>     3. **Equation punctuation**: Add proper punctuation to all equations
>
> ### Responses to Questions
>
> - **Q1: Relationship to multi-level hierarchies like HAC?**
>
>     HAC creates a fixed multi-level hierarchy where each level operates at progressively longer time scales (e.g., Level 0: 1-step actions, Level 1: 10-step subgoals, Level 2: 100-step subgoals). Each level sets goals for the level immediately below it, forming a vertical stack.
>
>     MRS differs fundamentally:
>
>     - **Single hierarchy level**: One manager, one worker (not multiple levels). All temporal resolutions exist in parallel to each other rather than in a vertical hierarchy.
>     - **Dynamic temporal selection**: The manager dynamically selects which temporal resolution to use based on the current state, rather than progressing through hierarchical levels.
>     - **Parallel skill experts**: Multiple resolution-specific skill heads are trained simultaneously; the choice head learns when each resolution is appropriate.
>     - **Joint optimization**: All components train with a single unified objective, rather than separate objectives per level.
>
>     The key insight is that **different temporal resolutions may be optimal in different states within the same task** (see Figure 5,7 and Section G), rather than different hierarchical levels handling different degrees of task abstraction. Our ablation studies (Figure 7) demonstrate that no single temporal resolution performs best across all tasks, motivating the need for dynamic selection.
>
>     We will expand our related work section to include a thorough discussion of HAC and other multi-level hierarchies, clearly positioning MRS as a complementary approach that provides temporal flexibility within a single hierarchy level.
>
> - **Q2: Comparison with HIQL and HAC**
>
>     Comparison with HIQL may be challenging as it is an offline approach and requires pre-collected data. We will try to provide a reasonable comparison if possible.
>
> - **Q3: Number of seeds**
>
>     We are a bit constrained on academic compute resources right now. But we commit to increasing the seeds for the final version.
>
> We believe these additions will significantly strengthen the paper and address all major concerns. We appreciate the reviewer's feedback and the opportunity to improve our work.

---

### Official Review · Reviewer_MhJ4 · 2025-11-01

**Soundness:** 3
**Presentation:** 3
**Contribution:** 3
**Rating:** 6
**Confidence:** 4

**Summary:**

This work proposes Multi-Resolution Skills (MRS), a hierarchical reinforcement learning framework that generates plans at multiple temporal resolutions. It is composed of two components: manager and worker. The worker is trained to reach goal state s_{t+l}. The manager is trained to generate the goals for different temporal steps (l) and selecting among these resolution-specific goals based on the task. Empirical results show that MRS outperforms single-resolution baselines on long-horizon navigation tasks and achieves performance comparable to non-hierarchical state-of-the-art agents on standard continuous-control benchmarks. Ablation studies confirm that the performance gains is indeed from the multi-resolution design.

**Strengths:**

* The manager and worker model are trained with shared layers (all except the resolution specific layer) to minimize the increase in model size.
* The framework uses CVAE so the predicted goal in future state is conditioned on the current state, this is to try and constraint the goal to to achievable futures.
* The framework includes an escape/reset mechanism that is a ∞-horizon skill that is designed to be invoked in unusual or recovery states (e.g. after falling).

**Weaknesses:**

* The ablation study can benefit from including experiments comparing workers trained separate (i.e. an individual model for each temporal resolution) to show that the design choice of having shared layers do not significantly negatively impact the performance.
* The work uses a single ELBO across horizons, but do not show clearly whether the decoder would be blurring for larger time step. If reconstruction is poor, the “goal” may be a fuzzy or unrealistic state, which would weaken the whole “constrain to achievable futures” story
* CVAE is trained online with data from the current worker, so it only recalls what the current worker can already do. That makes skills conservative. This means “skill = abstract action” is actually “skill = whatever the current policy happened to do.” That’s narrower than the claims.
* Compute/reporting transparency could be improved. They say experiments take “2 days … 5M steps on an RTX 5000” (App. B), but not for baselines. So we can’t tell if MRS is actually more sample-efficient or just more update-efficient.

**Questions:**

* In figure 2, the symbol for output of the decoder is s_*. Why not use the same notation \hat{s}_{t + l} as in figure 3 a)?
* In figure 4, there seem to be an issue with the y axis on the quadruped run, pendulum swingup, cheetah run graph (shows .000 where it is supposed to be 1000) and on the push, puck n place graph (shows .100 where it is supposed to be -100).

---

> ### Author Response · Authors · 2025-11-21
>
> We thank the reviewer for their thorough review and accurate summary of our work. We appreciate the recognition of our core contributions: parameter-efficient multi-resolution design, CVAE-based achievable goal constraints, and the recovery mechanism. We address each concern below.
>
> 1. **Ablation on separate vs shared layers**
>
>     Excellent suggestion. We will add this experiment and update through a comment soon. We are currently working on this. We will report: (1) parameter count increase, (2) performance comparison, and (3) training time. We note that our design rationale for shared layers was two-fold: (a) maintain parameter efficiency for fair comparison, and (b) enable representation transfer across temporal resolutions, which we hypothesize improves generalization.
>
> 2. **Decoder blurring at larger time steps**
>
>     We agree this is an important concern and provide both qualitative and mechanistic evidence that blurring does not undermine our approach:
>
>     **Qualitative evidence:** The sample subgoals in Appendix F (Figures 10-13) show predictions for each temporal resolution without cherry-picking. Even 64-step predictions remain coherent and physically plausible, not "fuzzy blurs."
>
>     **Exploration mechanism as self-correction:** If a state representation $s_{t+l}$ has poor reconstruction quality, our exploratory objective (Eq. 5) explicitly assigns a higher reward to visiting such state transitions, encouraging the agent to gather more data until the CVAE learns to reconstruct them accurately. This creates a self-improving cycle: unclear predictions → exploration incentive → better data → improved CVAE → clearer predictions. At convergence, the CVAE should model all visited transitions accurately, limited only by model capacity and the environment's stochasticity.
>
> 3. **CVAE conservatism - "skills = current policy behavior”**
>
>     This is a fair observation about the relationship between learned skills and policy capabilities. We clarify:
>
>     **Exploration drives skill discovery:** Our exploratory objective (Eq. 5) actively seeks out state transitions that are poorly modeled by the CVAE. Therefore, the agent is incentivized to discover novel transitions rather than just repeat existing behaviors. At optimality under pure exploration, the CVAE should be able to recall any state transition accessible in the environment, limited only by model capacity.
>
>     **Task rewards narrow the scope by design:** When combined with task rewards, the agent naturally focuses exploration around task-relevant regions. This is intentional, as we want skills that are useful for the task, not maximally diverse but arbitrary behaviors. The degree of exploration vs. exploitation is controlled by the reward weighting ([1.0, 0.1] in our experiments).
>
>     **Evidence of diverse skill discovery:**
>
>     - In sparse-reward settings (Robotics Push/Pick, AntMaze in Figures 4 & 6) where task reward provides minimal guidance, the agent successfully learns complex skills.
>     - When trained purely on exploratory rewards (Appendix C, Figure 9), agents discover diverse, sophisticated behaviors including front flips, back flips, jumps, and headstands; demonstrating that the CVAE framework can capture rich behavioral repertoires when given exploration opportunity.
>     - In our skill repurposing experiments (Figure 8), MRS with exploration-discovered skills outperforms diversity-maximizing approaches (DIAYN, ReST) on most tasks, suggesting that our abstract state-transition-based skills are more useful than "diverse but arbitrary" ones.
> 4. **Compute transparency**
>
>     We apologize for the incomplete reporting. By efficiency, we meant comparing the number of calls to the train function required for each method. All methods take about the same wall-clock time. We added the line “2 days … 5M steps on an RTX 5000” in the appendix for reproducibility, not for comparison.
>
>
> ### Responses to Questions
>
> - Notation consistency ($s^*$ vs $\hat{s}_{t+l}$)
>
>     Thank you for catching this inconsistency. We will standardize to $\hat{s}_{t+l}$ throughout for clarity.
>
> - Y-axis formatting issues in Figure 4
>
>     This is a matplotlib rendering issue with scientific notation that clipped the left side of the image slightly. We commit to fixing all such issues.
>
>
> We believe these clarifications and the forthcoming ablation study address the main concerns. We will update this response with the separate vs. shared layers ablation results within the next few days and incorporate all fixes in the revised manuscript.

---

### Meta-Review · Area_Chair_MQTL · 2026-01-07

**Summary:**

The paper proposes Multi-Resolution Skills (MRS), a hierarchical RL method that extends the Director architecture by introducing multiple CVAEs to predict subgoals at varying temporal horizons. The method is evaluated on DeepMind Control Suite and sparse-reward navigation tasks (AntMaze), aiming to demonstrate that explicit temporal partitioning is a beneficial inductive bias for HRL.

The review process was mixed, leaning negative. While reviewers acknowledged the intuitive appeal of the multi-resolution idea and its potential for long-horizon tasks, the submission suffered from significant issues regarding statistical rigor and presentation quality. A primary concern was the use of only 3 seeds for experiments, which is insufficient for establishing reliable claims in high-variance HRL benchmarks. Furthermore, the paper was criticized for formatting errors and formality of language, as well as a lack of comprehensive baselines (e.g., HAC, HIQL). Although the authors promised to address these in the final version, the current state of the evidence does not meet the bar for acceptance.

**Reviewer Concerns:**

- The authors clarified the distinction between their method (parallel temporal resolutions) and hierarchical stacks like HAC, and provided reasoning for why Director was the primary baseline. They also addressed questions regarding the CVAE's ability to handle domain shifts during online training.

- The statistical weakness remains a major flaw. Reviewer PbWS correctly noted that 3 seeds are inadequate for empirical RL papers at this level. The authors' promise to run 7 seeds for the final version cannot be evaluated during the review process.

- The presentation quality was heavily criticized (Reviewer PbWS, XKvq) for numerous formatting errors and informal language. While fixable, it reflects a rushed submission.

- Incremental Novelty. Reviewer eWQ5 viewed the method as a straightforward multi-head extension of Director without deep theoretical justification or optimality guarantees, limiting its contribution compared to established theoretical frameworks.

The first issue is mostly resolved while the other issues remain outstanding.

**Reviewer Scores:**

All reviewers' ratings are unlikely to change.

Reviewer PbWS's primary objection was the lack of statistical rigor (3 seeds), which was not rectified during the rebuttal window.

Reviewer eWQ5 maintained a negative rating even after the rebuttal, citing the "marginal incremental" nature of the novelty and lack of theoretical grounding.

As for Reviewers MhJ4, XKvq, while they saw merit in the idea, the execution flaws prevented stronger support.

---

### Decision · Program_Chairs · 2026-01-26

Reject